# Acidic pH can attenuate immune killing through inactivation of perforin

Adrian W Hodel [1]✉, Jesse A Rudd-Schmidt [1,7], Tahereh Noori [1,7], Christopher J Lupton[2], Veronica C T Cheuk [1], Joseph A Trapani[3,4], Bart W Hoogenboom [5,6] & Ilia Voskoboinik [1,4]✉

## Abstract

**Cytotoxic lymphocytes are crucial to our immune system, primarily eliminating virus-infected or cancerous cells via perforin/granzyme killing. Perforin forms transmembrane pores in the plasma membrane, allowing granzymes to enter the target cell cytosol and trigger apoptosis. The prowess of cytotoxic lymphocytes to efficiently eradicate target cells has been widely harnessed in immunotherapies against haematological cancers. Despite efforts to achieve a similar outcome against solid tumours, the immunosuppressive and acidic tumour microenvironment poses a persistent obstacle. Using different types of effector cells, including therapeutically relevant anti-CD19 CAR T cells, we demonstrate that the acidic pH typically found in solid tumours hinders the efficacy of immune therapies by impeding perforin pore formation within the immunological synapse. A nanometre-scale study of purified recombinant perforin undergoing oligomerization reveals that pore formation is inhibited specifically by preventing the formation of a transmembrane β-barrel. The absence of perforin pore formation directly prevents target cell death. This finding uncovers a novel layer of immune effector inhibition that must be considered in the development of effective immunotherapies for solid tumours.**

**Keywords** CAR T; Cytotoxic T Lymphocytes; Immunotherapy; Tumour Microenvironment
**Subject Categories** Cancer; Immunology; Membranes & Trafficking

## Introduction

Cytotoxic lymphocytes, encompassing cytotoxic T lymphocytes (CTLs) and natural killer (NK) cells, are immune effectors that rely on the pore-forming protein perforin and pro-apoptotic serine proteases, granzymes, to kill virus-infected or transformed/pre-cancerous cells (Thiery and Lieberman, 2014; Gilbert et al, 2013; Voskoboinik et al, 2015). Activated CTLs recognise their target and form an immunological synapse. Intracellular cytotoxic granules that store perforin and granzymes fuse with the plasma membrane, releasing their contents into the synaptic cleft. Once released, perforin forms large transmembrane pores in the plasma membrane of target cells (Lopez et al, 2013a), enabling diffusion of granzymes into the target cytosol where they trigger various apoptotic pathways (Lopez et al, 2013b).

In a common form of anti-cancer immunotherapy, T lymphocytes are genetically modified to express a chimeric antigen receptor (CAR), transforming them into CAR T cells that, following ex vivo expansion and re-infusion into the patient, direct their powerful cytotoxicity against cancerous cells. Currently, CAR T immunotherapies are successfully used against some haematological malignancies (Cappell and Kochenderfer, 2023), but are largely ineffective against any solid tumours (Marofi et al, 2021; Sterner and Sterner, 2021; Raskov et al, 2021). This is attributed to the immunosuppressive microenvironment of solid tumours (Labani-Motlagh et al, 2020; Giraldo et al, 2018), prompting significant efforts to render it more permissive to immunotherapy. One aspect of immunosuppression that has received limited attention is the acidic environment of solid tumours. Tumour acidification occurs due to uptake of glucose and release of lactic acid into the tumour microenvironment as the preferred metabolic pathway (Kato et al, 2013), and reduced pH has been shown to directly affect immune killing (Nakagawa et al, 2015; Vuillefroy de Silly et al, 2024). Advances in magnetic resonance imaging using pH-sensitive contrast agents provided visualisations of heterogeneous pH distributions in vivo in the extracellular milieu of solid tumours in mice (Liu et al, 2012; Delli Castelli et al, 2014; Anemone et al, 2017, 2021; Gallagher et al, 2008) and humans (Jones et al, 2017), with pH frequently found between 6 and 6.5. Intratumoural pH below 6.5 has also been shown with a recently developed pH-sensitive fluorescent cell surface marker (Rohani et al, 2019) and an in vivo litmus test in human lymphoma (Miripour et al, 2020). Intriguingly, a recent study finds polarised areas around individual cancer cells reaching pH 5.3 (Feng et al, 2024).

The pore-forming activity of perforin is exquisitely pH-dependent, being most potent at neutral pH. Under these conditions, perforin monomers bind the lipid membrane in a calcium-dependent fashion (Uellner et al, 1997; Yagi et al, 2015;

[1]Killer Cell Biology Laboratory, Peter MacCallum Cancer Centre, Melbourne, VIC, Australia. [2]Biomedicine Discovery Institute, Department of Biochemistry and Molecular Biology, Monash University, Melbourne, VIC, Australia. [3]Cancer Cell Death Laboratory, Cancer Immunology Program, Peter MacCallum Cancer Centre, Melbourne, VIC, Australia. [4]Sir Peter MacCallum Department of Oncology, University of Melbourne, Melbourne, VIC, Australia. [5]London Centre for Nanotechnology, University College London, London, UK. [6]Department of Physics and Astronomy, University College London, London, UK. [7]These authors contributed equally: Jesse A Rudd-Schmidt, Tahereh Noori.
✉E-mail: adrian.hodel@petermac.org; ilia.voskoboinik@petermac.org

Voskoboinik et al, 2005). The monomers then assemble into small oligomers containing two to eight protein subunits. As these oligomers are not yet inserted into the membrane, they are referred to as prepore assemblies or 'prepores' (Leung et al, 2017). Initiated by their assembly and through a transition within the protein structure, prepores insert into the membrane and act as a nucleation site for further, rapid prepore binding and insertion, thereby growing the pore diameter. Mature perforin pores are heterogeneous in diameter and curvature and can form arc- or ring-shaped structures (Metkar et al, 2014; Leung et al, 2017; Rudd-Schmidt et al, 2019). In general, they contain 19–24 subunits and span lumen diameters of 10–20 nm (Law et al, 2010a; Ivanova et al, 2022). This process is greatly perturbed under acidic conditions: perforin lysis becomes reduced below pH 6.6 and is fully lost at pH 6 for both mouse (Young et al, 1987; Kataoka et al, 1997; Voskoboinik et al, 2005) and human perforin (Praper et al, 2010). At such low pH, calcium-dependent binding of perforin to lipid membranes remains intact, but the structural transitions necessary to form mature pores are blocked. Despite this, perforin is not denatured by the acidic pH and regains its lytic function upon restoration of pH levels to 7.4 (Praper et al, 2010; Lopez et al, 2013b).

We hypothesized that inhibition of perforin pore formation at the acidic pH observed in solid tumours should limit immune killing. However, it has never been determined to what extent extracellular pH can affect perforin function within a physiological immunological synapse. Using different types of cytotoxic lymphocytes, including anti-CD19 CAR T cells, in a range of immunological assays, we assessed how immune killing is influenced by pH. In addition, we used nanoscale imaging techniques to identify how perforin function is affected at acidic pH at a single-molecule level.

## Results and discussion

### Effector killing is attenuated at acidic pH despite significant release of perforin

To test the killing capacity of primary human immune effector cells, we used anti-CD19 CAR T cells that can readily recognise and kill CD19-positive target cells. As target cells, we transduced the U937 cell line, which exhibits a high sensitivity to perforin-mediated immune killing, to express the extracellular domain of CD19 (hCD19t). To conduct assays at neutral pH 7.4 and acidic pH 6, we developed a pH-stable cell media that does not require equilibration with $CO_2$. To achieve this, we equilibrated bicarbonate-free DMEM with two buffers, 20 mM HEPES (pKa 7.3, 37 °C) and 20 mM MES (pKa 6, 37 °C) (Appendix Fig. S1).

To measure pH-dependent immune killing, $^{51}Cr$-labelled U937-hCD19t cells were resuspended with anti-CD19 CAR T cells at different effector-to-target cell ratios and at various pH. The release of $^{51}Cr$ from target cells after 4 h, where predominantly perforin/granzyme-mediated immune killing occurs, was gradually reduced with decreasing pH until it was abrogated at pH 6 (Fig. 1A). The 4 h exposure to the acidic pH did not influence effector cell viability, and immune killing was fully restored after resuspending the cells to neutral pH (Fig. 1B).

Since efficient CAR T cell killing requires sequential antigen binding, immune synapse formation and degranulation, perforin pore formation, and granzyme-induced apoptosis, we reasoned that the disruption of any of these events would detriment CAR T cell cytotoxicity. Therefore, we first tested degranulation at acidic pH using externalisation of the granule-associated protein CD107a (LAMP1) (Betts and Koup, 2004; Alter et al, 2004). We found that CAR T cell degranulation was progressively reduced, but not abolished by lowering media pH, with 27% of effector cells degranulating at pH 6, compared to 65% at pH 7.4 (58% reduction) (Fig. 1C), and further infer that target cell recognition and synapse formation remain at least partially intact at pH 6. Similar observations were made with the natural killer cell line NK-92, where no killing was observed at pH 6, despite appreciable degranulation still being present (32% at pH 6 compared to 61% at pH 7.4; 48% reduction) (Fig. EV1). To adequately interpret this apparent disparity between immune killing and degranulation, we next assessed whether a similar reduction in degranulation is sufficient to abrogate immune killing under more physiological conditions. To this end, we analysed recently published data on the effect of hypomorphic disease-causing mutations in two proteins responsible for cytotoxic granule exocytosis, MUNC13D and STXBP2, on CTL degranulation and cytotoxicity (Noori et al, 2023). As expected, the degranulation and cytotoxicity of mutant CTLs were closely associated (Fig. 1D). This contrasted with degranulation and cytotoxicity of CAR T and NK-92 cells at acidic pH, which exhibit a sharp drop in killing below pH 6.3 without a corresponding reduction in degranulation (Fig. 1E,F).

We next evaluated whether reduced killing at acidic pH reflected reduced perforin release. Using a recently established experimental system (Castiblanco et al, 2022; Rudd-Schmidt et al, 2022), we directly visualised ALFA-tag (Götzke et al, 2019) perforin (ALFA-PRF) secretion by murine CTLs at pH 7.4 or pH 6.0, using an artificial immunological synapse formed between the CTL and glass coverslips coated with anti-CD3/CD28 antibodies. Fluorophore-conjugated anti-ALFA-tag nanobodies were added to the culture media to visualise the secreted protein, and the accumulating fluorescence within the artificial synapse was quantified as shown in Fig. 2A,B. Consistent with degranulation assays of CAR T effectors, we observed ~50% reduced but still significant secretion of ALFA-PRF at pH 6.

Lastly, we assessed whether released perforin could remain sequestered by the cytotoxic granule storage protein serglycin. The interaction between serglycin and perforin is both physiologically important (Sutton et al, 2016) and pH-dependent (Masson et al, 1990). Serglycin sequesters and thereby inactivates perforin at the low pH in cytotoxic granules during storage and, after secretion, dissociates at the neutral pH of the immunological synapse. To assess if serglycin plays a role in attenuating immune killing at acidic pH, we tested whether serglycin deficiency improves immune killing in OTI CTLs. Having confirmed that the acidic pH had no effect on SIINFEKL antigen presentation by EL4 target cells (Fig. 2C,D), and that target cell killing by transgenic OTI CTLs recovered at pH 7.4 after a 4 h exposure to the media at pH 6 (Fig. 2E), we then determined the activity of wild-type and serglycin-deficient OTI CTLs at different pH (Fig. 2F). We found that the two cell types responded to the changing pH almost identically (Fig. 2G) suggesting that serglycin plays no role in attenuating immune killing at acidic pH.

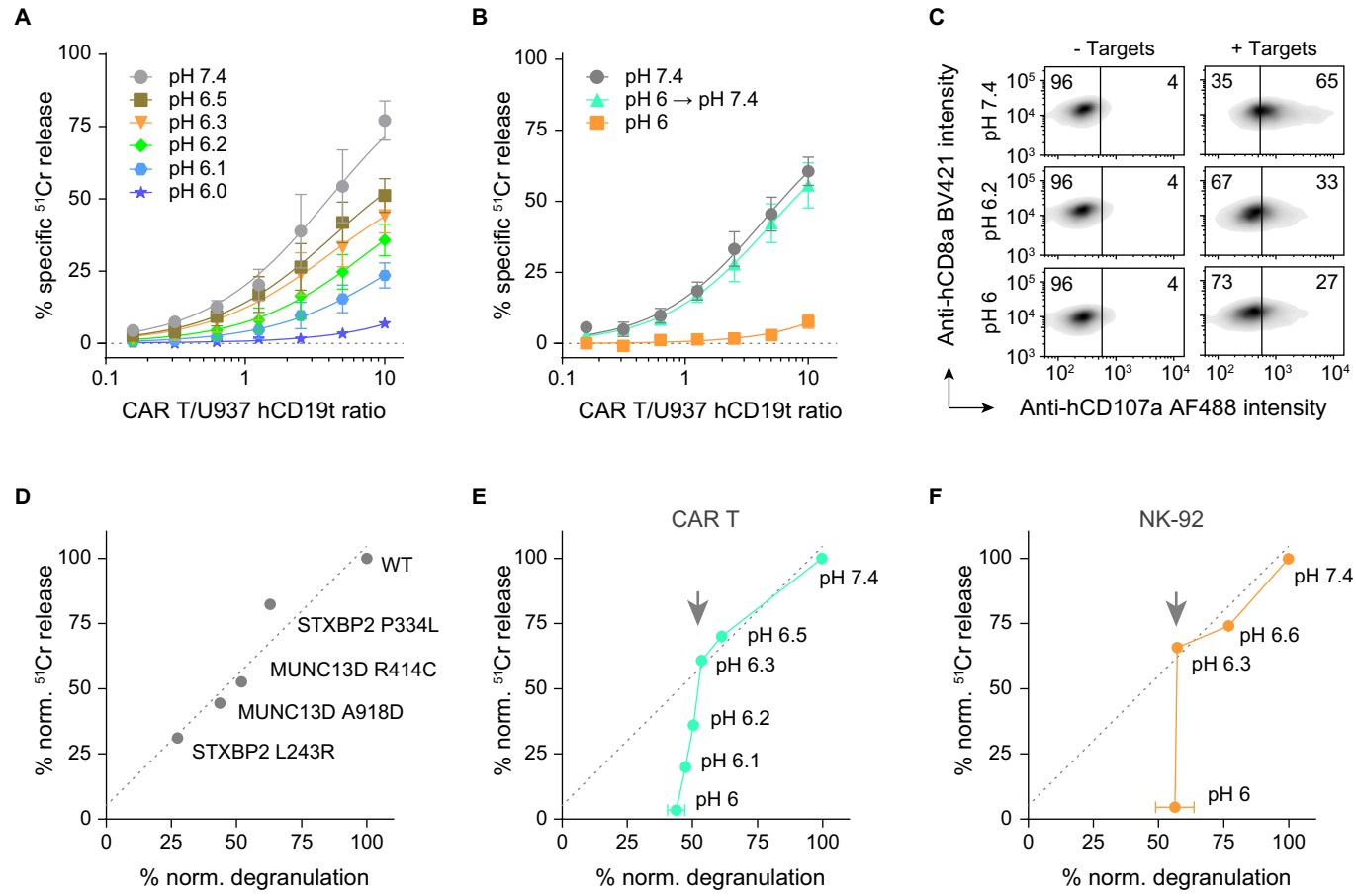

**Figure 1. pH-dependent immune killing and degranulation of different immune effectors.**

(A) Anti-hCD19 CAR T effectors killing U937 hCD19t targets at different pH, shown as a function of effector-to-target cell ratios, and measured by the release of $^{51}$Cr radionuclide label. (B) Rescued immune killing of effectors that have previously been incubated at pH 6 for 4 h at 37 °C and were then resuspended at neutral pH and used in a killing assay immediately afterwards. Controls at pH 7.4 and pH 6 were using immune effectors without prior incubation at acidic pH. (C) Flow cytometry-based detection of degranulation of anti-hCD19 CAR T effectors in the absence ('− Targets') and presence ('+ Targets') of U937 hCD19t targets at neutral and acidic pH. Samples without targets were used to determine the cut-off for degranulation. Percentages of the CAR T cell population positive or negative for degranulation are indicated by the numbers in the upper corners. (D) Graph correlating degranulation and immune killing at neutral pH based on different hypomorphic mutations of granule trafficking proteins MUNC13D and STXBP2 obtained in a different study (Noori et al, 2023). Degranulation and $^{51}$Cr release were normalised to WT (100%). (E) Equivalent to (D) but using data from anti-hCD19 CAR T killing and degranulation at different pH as shown in (A, C). The line fit is carried over from (D). The arrow indicates the pH below which the data curve deviates from the line fit. (F) As in (D, E) using NK-92 effectors against K562 target cells. Data information: (A, B) depicts mean ± SE from $n = 3$ biological replicates, fitted with Michaelis–Menten kinetics. In (D), $^{51}$Cr immune killing was calculated from Michaelis–Menten fits of mean values from $n = 3$ biological replicates at a 1:1 effector-to-target ratio and plotted against mean degranulation of $n = 2$ biological replicates. (E, F) show mean ± SD of $n = 3$ biological replicates (pH 7.4, pH 6) or a single experiment (pH 6.6–6.1). Source data are available online for this figure.

## pH-dependent immune killing correlates with the pore-forming activity of perforin

As our live-cell data supported the idea that acidic extracellular pH affects perforin function after its release into the immunological synapse, we assessed the molecular basis of pH-dependent pore formation by perforin. To this end, we used purified recombinant murine wild-type perforin (WT-PRF) in conjunction with turbidity-based sheep red blood cell (SRBC) lysis assays (Appendix Fig. S2A,B) and atomic force microscopy (AFM) on solid-supported phospholipid bilayers. MMT buffer (see 'Methods') had no detectable effect on the lytic activity of WT-PRF compared to the commonly used buffer but controlled the experimental conditions at pH 4–9 (Appendix Fig. S2C). SRBCs were stable

overnight at pH ≥5.5 (Appendix Fig. S2D), and a supported lipid bilayer as used in AFM experiments remained stable at pH ≥ 4 for the duration of the experiments (~1 h).

We set up SRBC lysis assays at a range of WT-PRF concentrations and the pH range between 5.5–7.5. The change of turbidity that reflected SRBC lysis was monitored over 6 h at 37 °C. The results highlight that acidification slows cell lysis and concomitantly reduces the maximum achievable level of lysis (Fig. 3A). Extreme concentrations of WT-PRF were still able to lyse SRBCs at pH 6 (at reduced capacity), but not at 5.5. In addition, molecular scale images from AFM experiments (for practical reasons at a single WT-PRF concentration and not time-resolved) produced a marked reduction of WT-PRF arc- and ring-shaped pores at pH 6–6.5 and their complete absence at and below pH 5.5 (Fig. 3B). Quantification of WT-PRF pore

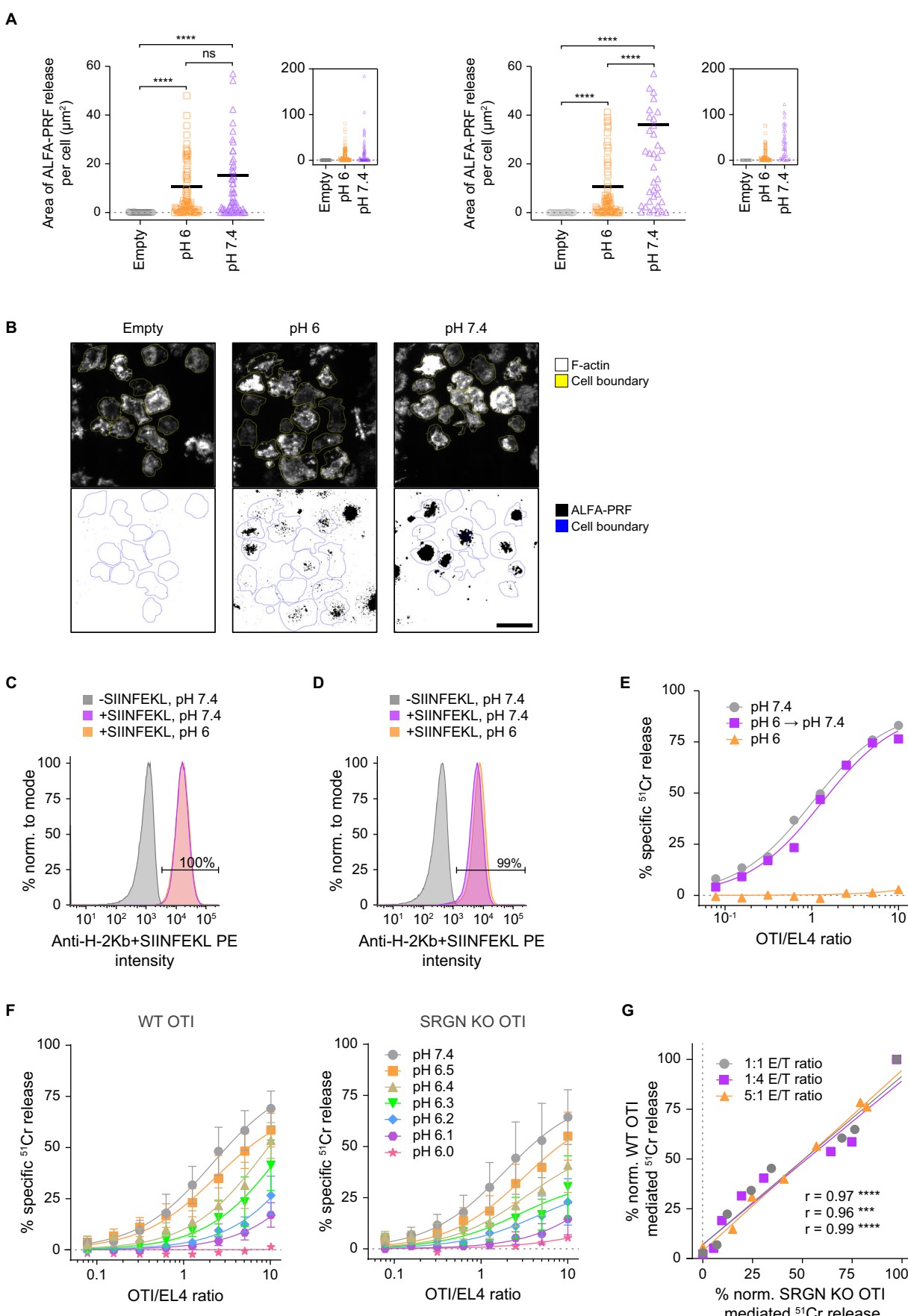

**Figure 2. Release of ALFA-PRF into artificial synapses at different pH.**

(A) Data showing the area of ALFA-PRF release per cell into the artificial synapse assessed after 45 min. Cells that were not expressing ALFA-PRF ('Empty', shown in grey) were used as negative control and compared to ALFA-PRF release at pH 6 (orange) and pH 7.4 (purple). Data is shown magnified (large plots) or at full range (insets). Horizontal lines depict means. (B) Representative fluorescence images of actin (top) and ALFA-PRF (bottom) displayed by effector cells engaged in artificial synapses. ALFA-PRF images are shown in binary after background subtraction. The coloured contours are manually traced cell boundaries used to calculate the area of ALFA-PRF release per cell. All experiments were blinded. (C) EL4 cells were pulsed with SIINFEKL at pH 7.4 and subsequently incubated at pH 6 for 4 h at 37 °C. The cells were then stained with anti-H-2Kb+SIINFEKL PE at neutral pH and their fluorescence measured by flow cytometry. (D) EL4 cells were pulsed with SIINFEKL at pH 6 and then stained with anti-H-2Kb+SIINFEKL PE at neutral pH. The percentage gate is shown for populations incubated at pH 6, compared to untreated EL4 cells. (E) OTI immune cells were incubated for 4 h in pH stable media at pH 6, 37 °C, 0% $CO_2$, after which the pH was neutralised and an immune killing assay was set up immediately, including controls for OTI effectors that have not been in acidic conditions prior, at pH 6 and pH 7.4. After neutralising pH, OTI effectors killed their EL4 targets at the same rate as without prior acidification. (F) WT and serglycin (SRGN) KO OTI CTL killing of SIINFEKL pulsed EL4 cells at different pH. (G) Plot of WT vs. SRGN KO OTI killing at indicated effector-to-target (E/T) ratios. Pearson's correlation coefficients (r) for the different ratios are shown in the bottom right corner in the same order as the legend. Data information: (A) shows individual and mean values (horizontal lines) from $n = 2$ biological replicates in separate plots. The significance levels are determined using the $P$ value from Kruskal–Wallis and uncorrected Dunn's post hoc tests, ****$P < 10^{-4}$, ns, not significant, $P = 0.76$. (B), scale bar, 20 μm. (E) shows mean values from $n = 2$ biological replicates, fitted with Michaelis–Menten kinetics. (F) shows mean ± SE from $n = 3$ biological replicates, fitted with Michaelis–Menten kinetics. (G) [51]Cr release was calculated according to data fits shown in (F). Data were analysed using Pearson's correlation with two-tailed $t$ test, ***$P = 10^{-4}$, ****$P < 10^{-4}$. Source data are available online for this figure.

formation and subsequent fit with a four-parameter logistic curve yielded a half maximum inhibition at pH 6.4 ± 0.2 (mean ± 95% confidence interval) (Fig. 3C, black open circles). These findings were remarkably consistent with SRBC lysis data (Fig. 3C, orange triangles), and immune killing (Fig. 3C, blue squares), directly linking the attenuation of immune killing to reduced WT-PRF pore formation at acidic pH.

WT-PRF function in the presence of SRBCs and $Ca^{2+}$ ions was largely restored, even after hours-long incubation at pH 5.5, by neutralising the pH (Fig. 3D). Similarly, AFM images show recovery of arc- and ring-shaped pore assemblies after neutralising the pH in samples that initially exhibited few or no pores (Fig. EV2A). The size distribution of these assemblies represents an endpoint characteristic for a pore-forming pathway. We therefore traced perforin assemblies in the collected AFM data (see Methods). Pores formed at pH 6.5 and pH 6, though lower in number, show arc- and ring-shaped pores similar in size to those formed at pH 7.4, indicating that they were formed via the same pore-forming pathway (Fig. EV2B). We then evaluated the assemblies found on samples that have been recovered from acidic pH. We first note that the protein density on the sample surface is conserved across all pH levels despite removing unbound protein when recovering pH (Fig. EV2C). Furthermore, the size distributions were conserved across all pH levels (Fig. EV2D). Taken together, these observations indicate normal binding of WT-PRF to the lipid membrane at acidic pH and, after the pH was restored to neutral, the protein recommenced its natural pore-forming pathway. Following the already well-established mechanism of perforin pore formation (Leung et al, 2017), we went on to explore how acidic pH inhibits the perforin pore.

## Perforin remains in prepore-like assemblies at acidic pH

Perforin is released into the extracellular space as soluble monomer. Using mass photometry (Young et al, 2018) to measure the molecular weight of WT-PRF in solution (Fig. EV3A,B), we found that it exists as a monomer at neutral pH but, at pH 5.5, ~70% of the protein forms dimers and ~20% remains in a monomeric form, with the rest of the protein found in higher-order assemblies (Fig. EV3C). Restoring the pH from 5.5 to 7.4 causes dissociation back into monomeric protein (Fig. EV3D).

We next assessed perforin binding to lipid membranes at acidic pH. Using AFM, we can detect superficially bound WT-PRF at pH 5 after trapping it into plaques by adding a crosslinker (Appendix Fig. S3A–C). The height of these plaques corresponds to the height of an upstanding perforin monomer, ~11 nm (Leung et al, 2017), in line with a membrane-proximal location of the lipid binding C2 domain of perforin (Law et al, 2010a; Yagi et al, 2015). Since C2 domain binding is calcium-dependent (Uellner et al, 1997; Voskoboi-nik et al, 2005), we further tested if this is also the case at acidic pH using SRBC binding assays in the presence or absence of 1 mM $Ca^{2+}$ at neutral pH or pH 5.5. To prevent cell lysis during the assays at neutral pH, we used a non-lytic perforin mutant with an engineered disulphide-lock, TMH1-PRF (Leung et al, 2017). The flow cytometry data (Fig. 4A) shows membrane binding of TMH1-PRF at pH 7.4 in the presence of $Ca^{2+}$ and some binding in the absence of $Ca^{2+}$. Presumably, the latter one does not involve the essential C2 domain, since the unlocked TMH1-PRF (by reduction of its engineered disulphide bond) has no haemolytic activity in the absence of $Ca^{2+}$ but is functional in the presence of $Ca^{2+}$ (Appendix Fig. S3D; Leung et al, 2017). At pH 5.5, TMH1-PRF binding is only detectable in the presence of $Ca^{2+}$ (Fig. 4B). To confirm this further, we found that the perforin C2 domain mutant D429A, which completely disrupts membrane binding at neutral pH (Voskoboinik et al, 2005; Yagi et al, 2015; Hodel et al, 2021), also does not bind to cells at pH 6 (Fig. 4C). Lastly, the amount of protein binding is similar at acidic and neutral pH of TMH1-PRF to SRBCs at pH 5.5 (Fig. 4A,B), and WT-PRF to K562 cells at pH 6 (Fig. 4D, tested on ice to prevent lysis). Taken together, this data suggests that perforin binds the lipid substrate at acidic pH through its C2 domain, as it would under neutral conditions.

Once perforin binds to the lipid membrane at neutral pH, it assembles into short, linear prepores that typically contain only two to eight protein subunits (Leung et al, 2017; Ivanova et al, 2022). They are thus much smaller than mature pores that typically contain more than 16 subunits and can close into a ring-shape. In addition, average spacing between subunits changes from 3.9 nm in prepores to 2.6 nm in pores (Leung et al, 2017). With this information, we sought to identify which part of the perforin assembly pathway was affected by acidic pH. We used a negative-stain electron microscopy dataset (Lopez et al, 2013b) of perforin bound to lipid monolayers at pH 6 and at neutral pH in MMT buffer (Fig. 4E) and quantified it as follows: the subunit spacing at

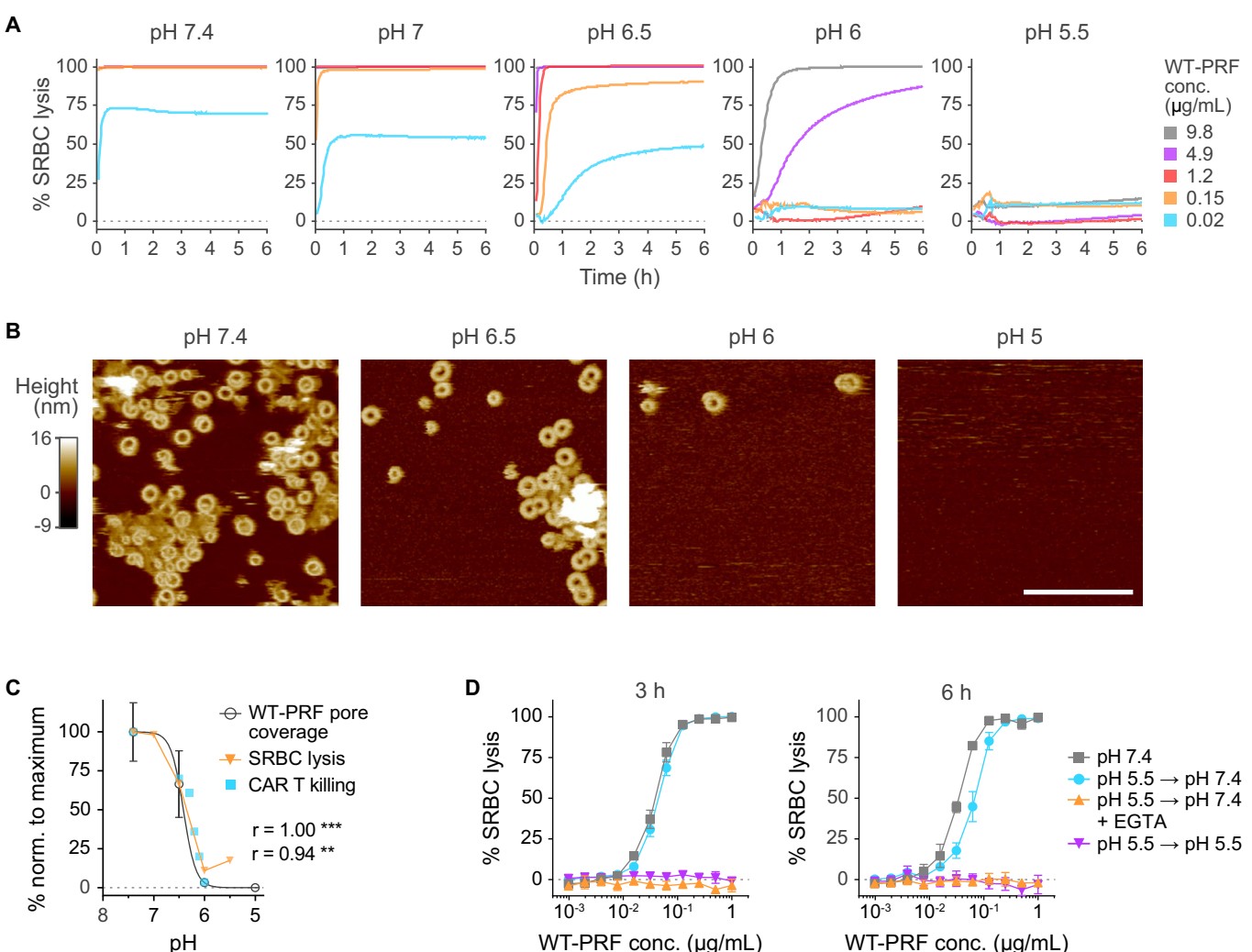

**Figure 3. WT-PRF mediated lysis and pore formation is reduced at acidic pH and can be restored.**

(A) 6 h timelapse of WT-PRF mediated lysis of SRBCs at pH 7.4–5.5 and five different protein concentrations at 37 °C. 1 μg/mL ≈16 nM WT-PRF. (B) Representative AFM images of supported lipid bilayers incubated with WT-PRF at the indicated pH. Perforin pores are visible as golden ring- and arc-shaped features, while the lipid background is coloured maroon. Pore formation is gradually reduced with pH, and below pH 6, no pore features were observed. The images are re-used from a larger dataset shown in Fig. EV2A. (C) Overlay of SRBC lysis shown in (A) at 0.15 μg/mL and 0.5 h, WT-PRF pore coverage quantified from AFM data as in (B), and CAR T cell immune killing at a 1:1 effector-to-target ratio calculated from the fits shown in Fig. 1A. All values were normalised to lysis, pore formation, and immune killing, respectively, observed at pH 7.4 (100%). Pearson's correlation coefficients ($r$) between WT-PRF pore coverage and SRBC lysis or CAR T killing data, respectively, are shown in the bottom right side of the graph. (D) Recovery of SRBC lysis after incubation at pH 5.5 and indicated incubation times (3 h or 6 h) when neutralising pH. The calcium chelator EGTA was added while restoring pH in control samples. Data information: (B) scale bar, 200 nm. In (C), WT-pore coverage shows mean ± SD, quantified from $n = 5$ AFM images, taken across the sample surface and covering a total area of 1.5 μm² each, and was fitted with a four-parameter logistic curve. The fitted values were used to perform Pearson's correlation analysis with two-tailed $t$ test, **$P = 5 \times 10^{-3}$, ***$P = 3 \times 10^{-4}$. (D) Shows mean ± SE from $n = 3$ technical replicates. Source data are available online for this figure.

pH 6 was measured at $4.0 \pm 0.9$ nm compared to $2.6 \pm 0.6$ nm (mean ± standard deviation) at pH 7.4 (Fig. 4F). Furthermore, the assemblies formed at pH 6 contain fewer than ten protein subunits throughout, in contrast to the larger arc- and ring-shaped assemblies formed at pH 7.4 (Fig. 4G). By comparing these results with the behaviour of perforin prepores and pores, we conclude that at acidic pH, perforin is unable to insert its transmembrane helices into the membrane to form a β-barrel pore and remains in a prepore-like state, as illustrated in Fig. 4H.

## Concluding remarks

In summary, we tested the effect of extracellular acidification on the killing ability of various types of cytotoxic lymphocytes - primary human CAR T cells, an NK cell line, and primary mouse CTLs. Despite having fundamentally different receptor-antigen interactions, they all showed similar sensitivity to acidic pH, resulting in the rapid decline of immune killing at pH lower than 6.5, and the loss of activity at pH 6.

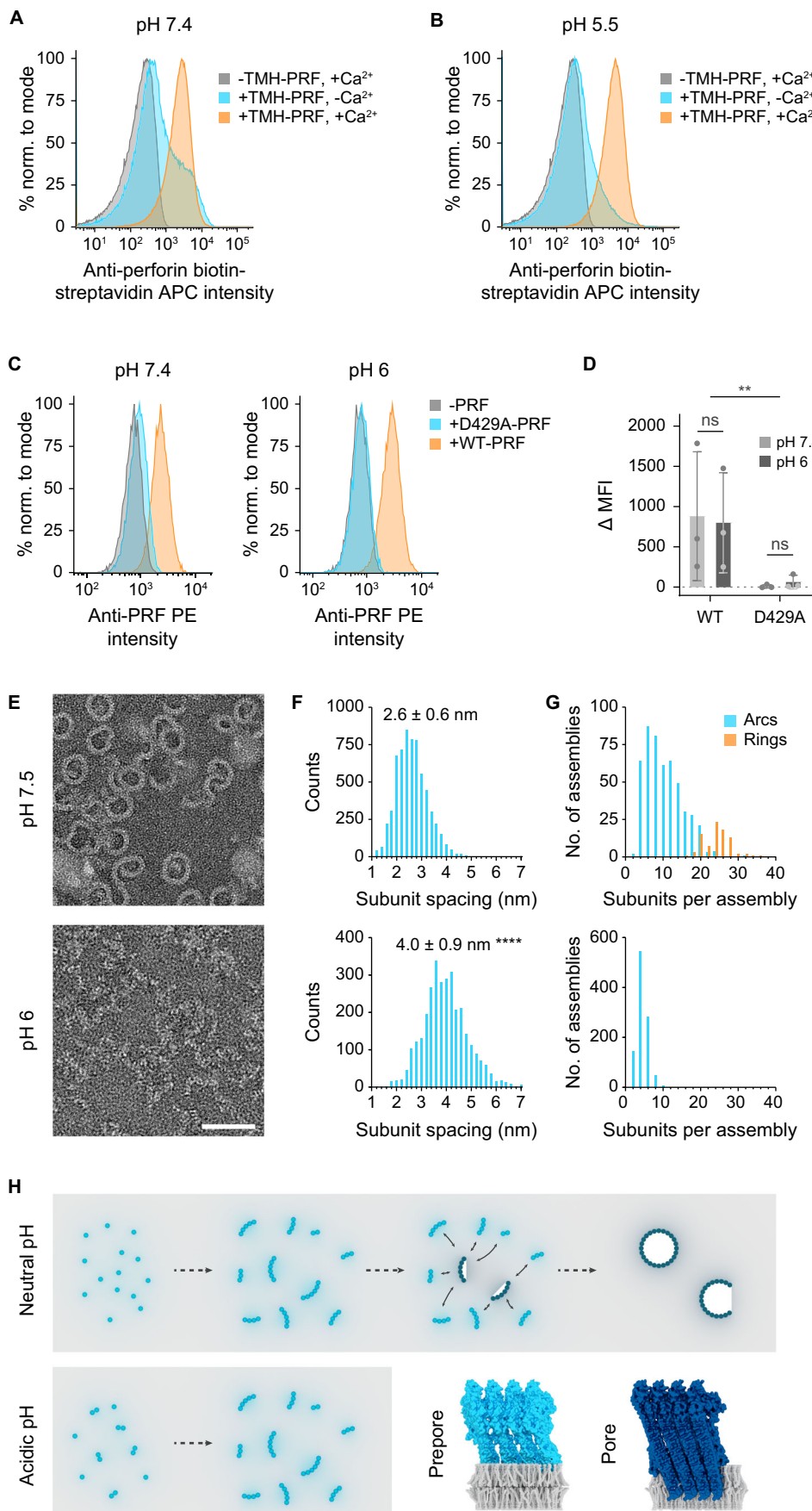

**Figure 4. Perforin binding remains intact, and EM images of perforin bound to lipid monolayers show prepore-like assemblies at acidic pH.**

(A, B) Binding of 1 µg/mL non-lytic perforin mutant, TMH1-PRF, in the presence and absence of 1 mM $Ca^{2+}$ and at pH 7.4 (A) and 5.5 (B) against an untreated control at pH 7.4, evaluated by flow cytometry. (C) K562 cells were treated with WT-PRF, D429A-PRF, or no protein (-PRF) at pH 7.4 (left plot) or pH 6 (right plot) and subsequently labelled with an anti-PRF antibody. Irrespective of pH, flow cytometry data shows increased fluorescence only in the presence of WT-PRF but not D429A-PRF, indicating that the C2 mutant is not binding K562 cells. Sufficient $Ca^{2+}$ for perforin binding is present in the cell medium. (D) Grouped bar plot showing the increase in geometric mean fluorescence intensity (Δ MFI) upon the addition of WT-PRF and D429A-PRF at pH 7.4 and pH 6. (E) EM images of lipid monolayers containing WT-PRF assemblies formed at the indicated pH levels. (F) Histograms of spacings between adjacent WT-PRF subunits extracted from EM images shown in (E). (G) Histograms of subunits per WT-PRF assemblies shown in (E). Ring- and arc-shaped assemblies are plotted separately in orange and blue, respectively. (H) Schematic illustration of the molecular assembly pathway of perforin pore formation at neutral pH, compared to acidic pH. The two schematics on the bottom right illustrate the concept of membrane surface-bound non-lytic prepores and membrane inserted, lytic pores. Perforin models were used from the RCSB protein database (rcsb.org), accessions 3NSJ and 7PAG (Law et al, 2010b; Data ref: Law et al, 2010b; Ivanova et al, 2022; Data ref: Ivanova et al, 2022a). Data information: In (D), data are presented as mean ± SD of $n = 3$ biological replicates and analysed using Kolmogorov–Smirnov tests, **$P = 2 \times 10^{-3}$, ns, not significant, $P = 0.6$ (D429A), $P = 1.0$ (WT). (E) scale bar, 50 nm. (F) mean ± SD of subunit spacing indicated atop each histogram was significantly different in Kolmogorov–Smirnov test of unbinned and $\chi^2$ test of binned data, both ****$P < 10^{-4}$ (two-tailed) as indicated in the lower histogram. Number of analysed subunits were $n \approx 6000$ (pH 7.4) and $n \approx 3000$ (pH 6). The numbers of analysed assemblies in (G) were $n = 495$ (pH 7.4) and $n = 1028$ (pH 6). A total sample area of 2.7 µm² was analysed for each pH in (F, G). Source data are available online for this figure.

Between pH 7.4 and pH 6.5, the attenuation of cytotoxicity correlates with the degree of lymphocyte degranulation. This was not surprising, given our own experimental data and other reports on degranulation inhibitors in CTLs (Tamzalit et al, 2020) and granule trafficking defects in immunodeficiencies (Gray et al, 2017). The finding is also in line with the pH sensitivity of ORAI1/STIM1 channels that control calcium influx and degranulation (Maul-Pavicic et al, 2011) and operate at approximately half capacity at pH 6.5 compared to neutral pH (Tsujikawa et al, 2015). However, below pH 6.5, immune killing was acutely reduced and then lost at pH 6. There, we also observed the reduction in degranulation, but this could not sufficiently explain the complete loss of function of cytotoxic lymphocytes. Instead, we found a near-perfect correlation between immune killing and the ability to of perforin to form transmembrane pores. Inactivation of perforin was thus the critical underlying cause of reduced immune killing below pH 6.5.

Considering the known structure of perforin (Ivanova et al, 2022; Law et al, 2010a) we developed a hypothesis for the structural basis of perforin inactivation at acidic pH. Perforin pore formation depends on the unfurling of two α-helical bundles per molecule, which results in two transmembrane β-hairpins. These hairpins connect with each other and with hairpins of adjacent subunits via hydrogen bonds, forming a stable β-barrel structure lining the pore (Ivanova et al, 2022). Intriguingly, the intra- and inter-subunit interface between hairpins of perforin is lined by seven (murine perforin) or eight (human perforin) histidine residues with a sidechain pKa ≈6 (Fig. EV4). Acidification would protonate these sidechains, reducing the number of possible hydrogen bonds between them. The functional implication of such weaker intermolecular binding would be a less efficient unfurling of the hairpins, leading to inefficient pore formation and protection of target cells against cytotoxic lymphocyte killing, as demonstrated in our data.

CTLs need to infiltrate tumours to exert their cytotoxic activity. Given the frequent observations of intratumoural pH levels well below 6.5 (as outlined in the Introduction), tumour infiltrating lymphocytes will encounter areas with sufficiently low pH to impair perforin function and cytotoxic lymphocyte killing in general. Particularly intriguing is the widespread occurrence of very low pH 5.3 directly at the plasma membrane of cancer cells (Feng et al, 2024), where perforin ought to unfold its pore-forming capacity. Although it is currently unclear to what extent immunological synapses are formed within these areas, reports of immune

co-receptor interactions involving VISTA specifically at pH ~6 (Yuan et al, 2021) support that effector-target interactions occur in such acidic environments.

The importance of our work is reflected by the persistently low efficacy of CAR T cell therapies against solid tumours. Within these tumours, the microenvironment acts as an unyielding barrier. Factors like checkpoint signalling and diminished cell metabolism (Labani-Motlagh et al, 2020; Giraldo et al, 2018; McPhedran et al, 2024; O'Sullivan and Pearce, 2015; Yin et al, 2019) hinder the long-term performance of immune effectors in eliminating their targets. However, potent immune killing ultimately hinges on perforin-mediated target cell death. Addressing perforin inactivation at the acidic intratumoural pH is thus necessary to improve clinical outcomes of immunotherapy.

The challenge of perforin inactivation at acidic pH can possibly be solved by raising the pH of the tumour microenvironment (Erra Díaz et al, 2018; Lacroix et al, 2018), which would also reduce other immunosuppressive effects. Alternatively, immune cells could be engineered to withstand acidity and maintain their cytotoxicity. To this end, our molecular analysis of how acidic pH impacts perforin provides a guide to the design of pH-insensitive perforin.

# Methods

**Reagents and tools table**

| Reagent/resource | Reference or source | Identifier or catalogue number |
|---|---|---|
| **Experimental models** | | |
| C57BL/6.OTI | The Jackson Laboratory | RRID: IMSR_JAX:003831 |
| EL4 | ATCC | TIB-39 |
| K562 | ATCC | CCL-243 |
| NK-92 | ATCC | CRL-2407 |
| U937 | ATCC | CRL-1593.2 |
| **Recombinant DNA** | | |
| MSCV IRES GFP | Addgege | Cat. No. 20672 RRID: Addgene_20672 |

| Reagent/resource | Reference or source | Identifier or catalogue number |
|---|---|---|
| **Antibodies** | | |
| Alpaca anti-ALFA clone 1G5 AZDye 568 | Nanotag Biotechnologies | Cat. No. N1502-AF568-L RRID: AB_3075980 |
| Goat anti-mIgG (polyclonal) PE | Life Technologies | Cat. No. A10543 RRID: AB_10374311 |
| Hamster anti-mCD28 clone 37.51 | BD Biosciences | Cat. No. 553295 RRID: AB_394764 |
| Hamster anti-mCD3ε clone 145-2C11 | BD Biosciences | Cat. No. 553058 RRID: AB_394591 |
| Mouse anti-H-2Kb+SIINFEKL clone 25-D1.16 PE | BioLegend | Cat. No. 141603 RRID: AB_10895905 |
| Mouse anti-hCD107a clone H4A3 AF488 | BioLegend | Cat. No. 328610 RRID: AB_1227504 |
| Mouse anti-hCD16 clone 3G8 BV711 | BioLegend | Cat. No. 302044 RRID: AB_2563802 |
| Mouse anti-hCD3 clone OKT3 | Invitrogen | Cat. No. 16-0037-85 RRID: AB_468854 |
| Mouse anti-hCD45r clone HI100 FITC | BD Biosciences | Cat. No. 555488 RRID: AB_395879 |
| Mouse anti-hCD54 clone HA58 PE | BD Biosciences | Cat. No. 555511 RRID: AB_395901 |
| Mouse anti-hCD56 clone 5.1H11 PE | BioLegend | Cat. No. 362508 RRID: AB_2563924 |
| Mouse anti-hCD8 clone SK1 BV421 | BioLegend | Cat. No. 344748 RRID: AB_2629583 |
| Mouse anti-hNKp44 clone 253415 APC | R&D Systems | Cat. No. FAB22491A |
| Mouse anti-hNKp44 clone 253415 PE | R&D Systems | Cat. No. FAB22491P |
| Mouse anti-myc clone 9B11 | Cell Signaling Tech. | Cat. No. 2276 |
| Mouse anti-perforin clone δG9 | BD Biosciences | Cat. No. 556434 RRID: AB_396418 |
| Mouse anti-perforin clone δG9 biotin | Ancell | Cat. No. 358-030 |
| Mouse anti-perforin clone δG9 PE | BioLegend | Cat. No. 308106 RRID: AB_314704 |
| Rat anti-flag clone L5 AF488 | BioLegend | Cat. No. 637318 RRID: AB_2728470 |
| Rat anti-mCD90.2 clone 53-2.1 FITC | BD Biosciences | Cat. No. 553004 RRID: AB_394542 |
| Streptavidin APC | eBioscience | Cat. No. 17-4317-82 |
| **Chemicals, enzymes, and other reagents** | | |
| 2-mercaptoethanol | Sigma-Aldrich | Cat. No. M3148 CAS No. 60-24-2 |
| 51Cr as sodium chromate | PerkinElmer | Cat. No. NEZ030005MC |
| Calcium chloride dihydrate | Sigma-Aldrich | Cat. No. C7902 CAS No. 10035-04-8 |
| Celpresol | Immulab | Cat. No. 06332301 |
| DL-malic acid | Sigma-Aldrich | Cat No. 240176 CAS No. 6915-15-7 |
| DMEM | Gibco | Cat. No. 11965-092 |
| DMEM, powder | Gibco | Cat. No. 12100-061 |

| Reagent/resource | Reference or source | Identifier or catalogue number |
|---|---|---|
| DOPC | Avanti Polar Lipids | Cat. No. 850375P |
| EGTA | Sigma-Aldrich | Cat. No. E3889 CAS No. 67-42-5 |
| Fatty acid-free bovine serum albumin | Roche Diagnostics | Cat. No. 10735086001 |
| GlutaMAX | Gibco | Cat. No. 35050-061 |
| Glutaraldehyde 8% | TAAB Laboratories Equipment | Cat. No. G010 |
| HEPES | Gibco | Cat. No. 15630-080 |
| Lentiboost | Revvity | Cat. No. SB-A-LF-901-01 |
| MEM Non-essential amino acids | Gibco | Cat. No. 11140-050 |
| MES hydrate | Sigma-Aldrich | Cat. No. M5287 CAS No. 1266615-59-1 |
| Penicillin-Streptomycin | Gibco | Cat. No. 15140-122 |
| Retronectin | Takara Bio | Cat. No. T100A |
| RPMI 1640 | Gibco | Cat. No. 11875-093 |
| SIINFEKL peptide | Genscript | Cat. No. RP10611 |
| Sodium Pyruvate | Gibco | Cat. No. 11360-070 |
| Tris | Merck Millipore | Cat. No. 1.08382.2500 CAS No. 77-86-1 |
| **Software** | | |
| Matlab R2018a | MathWorks | |
| NanoScope Analysis v2.0 | Bruker | |
| **Other** | | |
| μ-Slide 18 well 1.5H glass bottom chamber wells | Ibidi | Cat. No. 81817 |
| Cytation 3 | BioTek/Agilent Technologies | |
| Elyra PS1 | Zeiss | |
| MSNL probes for AFM | Bruker | Cat. No. MSNL-10 |
| MultiMode 8 AFM | Bruker | |
| TwoMP mass photometer | Refeyn | |

## Antibodies

Unless otherwise denoted, $2 \times 10^5$ nucleated cells or $\sim 2 \times 10^7$ sheep red blood cells (SRBCs) were stained in 50 μL of cell culture media or MMT buffer (pH 7.4) containing antibody diluted 1:100 for 30 min on ice. Primary/secondary stains were incubated sequentially, and the stained cells resuspended in 100–200 μL of cell culture media or buffer solution for flow cytometry.

## Cell culture

Primary human PBMCs and CAR T derivatives thereof were cultured in RPMI 1640 (Gibco, Life Technologies, Carlsbad, CA, USA) with 10% FCS, 2 mM Glutamax, 10 mM HEPES, 1 mM sodium pyruvate, 100 μM non-essential amino acids, 50 IU/mL

penicillin, 50 µg/mL streptomycin, and 100 IU/mL recombinant human IL2 (National Cancer Institute, Washington, Maryland, USA). To induce proliferation of CTLs after harvest, 30 ng/mL of anti-CD3 antibody clone OKT3 was added to the media once.

NK-92 cells were cultured in RPMI 1640 with 10% FCS, 2 mM Glutamax, 10 mM HEPES, 1 mM sodium pyruvate, 100 µM non-essential amino acids, 50 IU/mL penicillin, 50 µg/mL streptomycin, and 200 IU/mL recombinant human IL2.

Primary mouse splenocytes from C57BL/6.OTI transgenic mice were cultured in RPMI 1640 with 10% FCS, 2 mM Glutamax, 10 mM HEPES, 1 mM sodium pyruvate, 100 µM non-essential amino acids, 50 IU/mL penicillin, 50 µg/mL streptomycin, 50 µM 2-mercaptoethanol, and 100 IU/mL recombinant human IL2. To induce the proliferation of splenocytes after harvest, 10 ng/mL of SIINFEKL peptide were added to the media once. Murine experiments are approved by the Peter MacCallum Cancer Centre Animal Ethics Committee (AEEC) E655.

EL4 cells were cultured in DMEM (with high glucose, L-glutamine, and phenol red) with 10% FCS and 2 mM Glutamax and pulsed with 1 µg/mL SIINFEKL peptide (Genscript, Piscataway, NJ, USA) for 1 h at 37 °C prior to use as targets for OTI CTLs. U937 and K562 cells were cultured in RPMI 1640 with 10% FCS and 2 mM Glutamax.

Cells in RPMI 1640 were cultured at 37 °C, 5% $CO_2$, and cells in DMEM at 37 °C, 10% $CO_2$.

## Anti-hCD19 CAR T cells

CAR T effectors were generated from activated PBMCs isolated from healthy human donor blood. The cells were transduced with anti-hCD19 CAR using either retroviral or lentiviral transduction. For retroviral transductions, activated PBMCs were transduced with anti-hCD19 (clone FMC63) scFv_myc_hCD8hCD28_hCD3ζ CAR subcloned into a pSAMEN vector backbone using the supernatant of stably retrovirus producing PG13 cells and Retro-nectin (Takara Bio, Kusatsu, Shiga, Japan) coated culture plates as per the manufacturer's instructions. For lentiviral transductions, activated PBMCs were exposed to the viral supernatant of HEK293T cells transfected with anti-hCD19 (clone FMC63) scFv_flag_hCD8hCD28_hCD3ζ CAR subcloned into a pUltra vector backbone, and third generation lentivirus packaging vector in the presence of 0.25% v/v Lentiboost (Sirion Biotech, Gräfelfing, Germany). To obtain a pure population of anti-hCD19 CAR T cells, transduced PBMCs were sorted for CD8⁺/CD16⁻/myc⁺ or CD8⁺/CD16⁻/flag⁺ using fluorescently activated cell sorting (FACS) 6-7 days after harvest. CAR T cells were used up to 20 days after harvest. The FACS strategy is outlined in Appendix Fig. S4.

## U937 hCD19t target cell line

For hCD19 CAR T effectors to produce the highest readouts in degranulation assays, we empirically selected and designed the U937 hCD19t target cell line. To generate a sequence for human CD19t with truncated intracellular signalling domain (hCD19t), the open reading frame from the full-length hCD19 nucleotide sequence (GenBank NM_001770) was truncated at base pair 969, analogous to published work (Tey et al, 2007). The sequence was subcloned into an MSCV GFP vector backbone (Addgene 91975). HEK293T cells were transfected with hCD19t MSCV GFP and ampho retroviral packaging vector. The retroviral supernatant was used to transduce U937 cells in

Retronectin-coated plates as per manufacturer instructions. The transfected cells were sorted twice with respect to GFP reporter levels and anti-hCD19 antibody binding using FACS.

## pH stable cell media

pH-adjusted media for live cell assays was prepared from powdered, bicarbonate-free DMEM powder (Gibco) dissolved in Milli-Q water at the manufacturer-specified concentration and supplemented with 0.1% fatty acid-free bovine serum albumin (Roche Diagnostics, Mannheim, Germany), 2 mM Glutamax, 20 mM HEPES, and 20 mM MES hydrate (CAS 1266615-59-1, Sigma-Aldrich). The pH was subsequently adjusted to the desired levels at 37 °C by the addition of 1 M NaOH, monitored by a calibrated pH metre. Media toxicity was assessed as shown in Appendix Fig. S1. and was found to be comparable to standard T cell medium. pH stable cell media osmolarity was assessed by the Peter MacCallum Cancer Centre Media Kitchen facility and ranged between 300 and 320 mOsmol/kg, i.e., between T cell and standard DMEM culture media (Appendix Table S1).

For NK-92 cells, pH-sensitive experiments were carried out in RPMI 1640 with 10% FCS, 2 mM Glutamax, 10 mM HEPES, 1 mM sodium pyruvate, 100 µM non-essential amino acids, 50 IU/mL penicillin, 50 µg/mL streptomycin, 20 mM MES as buffer, besides sodium bicarbonate present in RPMI 1640. pH levels were adjusted by dropwise addition of 1 M malic acid at 37 °C. Note that this medium does not maintain pH values as stably as the pH-stable medium used for CAR T cells, and pH values may rise by up to 0.4 increments during the assay. Killing and degranulation assays using NK-92 effector cells against K562 target cells were otherwise carried out as with CAR T cells.

## pH stability of cell media

In total, 2 mL of different media was aliquoted into 10-mL centrifuge tubes. pH levels of media were measured before and after incubation for 4 h at 37 °C at levels of $CO_2$ and with or without effector/target cells as indicated. Where cells were present, effector cells were used at a density of $10^6$ cells/mL and target cells at $10^5$ cells/mL respectively, producing a 10:1 E/T ratio at the same cell densities used in immune killing assays.

## $^{51}$Cr release assays

As a measure of cell lysis, target cells were labelled with 50–100 µCi $^{51}$Cr radionuclide as sodium chromate in normal saline (PerkinElmer, Waltham, MA, USA) and incubated at 37 °C, 5% $CO_2$ for 1 h. Effector and target cells were washed and resuspended in pH-buffered media adjusted to the appropriate pH. A titration series of effector cells was combined with target cells in a 96-well plate containing $10^4$ targets and $10^5$ effectors in 200 µL media per well at the highest effector-to-target ratio. Wells containing either no effectors or 5% triton-X solvent solution were set up in parallel for each pH level as controls for spontaneous and total $^{51}$Cr release, respectively. The plates were incubated at 37 °C, 10% $CO_2$ for 4 h. The remaining cells were pelleted afterwards, and the supernatant extracted for γ-radiation decay counting on a Wizard2 Gamma Counter (PerkinElmer). The percentage of specific $^{51}$Cr release in a well $n$ was calculated using the radiation counts per minute (CPM):

Specific $^{51}$Cr release (%) = 100/(CPM$_{\text{total lysis}}$ − CPM$_{\text{spontaneous lysis}}$) ∗ (CPM$_n$ − CPM$_{\text{spontaneous lysis}}$). Michaelis–Menten kinetics were fitted with the maximum set to V ≤ 100% and the Michaelis constant set to k$_M$ > 0 as boundary conditions.

## Viability of effector cells at pH 6

To assess the viability of OTI/CAR-T effectors during our experimental conditions, $10^6$ effectors were stained with 200 µCi of $^{51}$Cr, washed, and resuspended at a concentration of $10^6$ cells/mL in 200 µL of bicarbonate-free DMEM, 20 mM HEPES, 20 mM MES, BSA or FCS as indicated, pH 7.4 and pH 6. For reference, mouse/human T-cell media at pH 7.4 or at pH 6 (acidified with HCl) with MES added as indicated was used. CAR-T effectors have additionally been exposed to U937 hCD19t effectors at a concentration of $10^6$ cells/mL (1:1 E/T ratio). After 4 h incubation at 37 °C and levels of CO$_2$ as indicated, cells were pelleted and 100 µL of supernatant extracted for radiation detection.

## Degranulation assays of anti-CD19 CAR T cells

Degranulation was measured by detecting CD107a on the surface of effector cells. To this end, $5 \times 10^4$ effector cells were mixed with $2 \times 10^5$ target cells in 200 µL of media and incubated for 3 h. Since antibody affinity changes with pH, all samples were washed three times with media at neutral pH before labelling in 50 µL of RPMI (pH 7.4) containing 1:100 anti-hCD8 BV421 and 1:25 anti-hCD107a AF488 antibodies for 30 min on ice. Samples were washed once after incubation and then resuspended in 100 µL of media for flow cytometry. Flow cytometry data for degranulation of effector cells was analysed as shown in Appendix Fig. S5A,B.

## Fluorescence microscopy of artificial immune synapses using ALFA-PRF OTI CTLs

Our experiments followed published protocols (Rudd-Schmidt et al, 2022). In brief, OTI CTLs were transduced with ALFA-PRF Tag-BFP MSCV and Lifeact-eGFP MSCV to visualise F-actin and sorted for Tag-BFP/eGFP double-positive cells using FACS. Ibidi µ-Slide 18 well 1.5H glass bottom chamber wells (Ibidi, Martinsried, Germany) were coated with 10 µg/ml anti-mCD3ε and 5 µg/ml anti-mCD28 antibodies in PBS overnight at 4 °C. The wells were washed with phosphate-buffered saline before use and equilibrated at 37 °C. Immediately prior to the experiment PBS was replaced by pH-stable media at pH 6 or pH 7.4 containing ~$10^5$ transduced OTI CTLs and 50 nM FluoTag-X2 anti-ALFA AZDye 568 nanobodies (Nanotag Biotechnologies, Göttingen, Germany). The release of ALFA-PRF over time was monitored with a Zeiss Elyra PS1 microscope equipped with an alpha Plan-Apochromat 100× oil lens (both Zeiss, Oberkochen, Germany) in total internal reflection fluorescence (TIRF) mode. In the recorded images, cell boundaries were traced manually using the F-actin signal to extract the area where ALFA-PRF is detected therein. Manual tracing was performed with samples blinded.

## Expression and purification of WT-PRF and TMH1-PRF

Recombinant wild-type mouse perforin (WT-PRF) and disulphide-locked mutant (A144C-W373C) perforin (TMH1-PRF) were expressed and purified using baculovirus infected Sf21 cells as described previously (Voskoboinik et al, 2004). The protein was eluted in 270 mM imidazole, 300 mM NaCl, 20 mM Tris, pH 7–8 at concentrations between 0.1 and 0.5 mg/mL.

## MMT buffer

Buffers were prepared in 0.9% saline solution (154 mM NaCl). To produce MMT buffered saline, 10 mM of DL-malic acid (CAS 6915-15-7, Sigma-Aldrich, St. Louis, MO, USA), 20 mM of MES hydrate, and 20 mM of Tris (CAS 77-86-1, Merck Millipore, Darmstadt, Germany) were added and adjusted between pH 4 and 8.5 using sodium hydroxide or hydrochloric acid. In all, 1–5 mM of CaCl$_2$·2H$_2$O (CAS 10035-04-8, Sigma-Aldrich) or 4 mM EGTA (CAS 67-42-5, Sigma-Aldrich) were added to buffers as outlined in the text.

## SRBC cell lysis timelapse assays

An SRBC stock was kept in Celpresol (Immulab, Parkville, VIC, Australia) at 4 °C. Prior to each assay, SRBCs were washed and resuspended at ~$2 \times 10^8$ cells/mL in MMT buffer adjusted to the desired pH levels and containing 2 mM Ca$^{2+}$. WT-PRF dilution series were set up on ice in 100 µL of buffer (without Ca$^{2+}$) and combined with 100 µL of SRBCs per well of a flat-bottom 96-well plate. Wells containing no WT-PRF were used as spontaneous lysis controls, and SRBCs lysed in Milli-Q water were used as total lysis controls. As a measure of lysis, turbidity was monitored in 2-min intervals at 600 nm absorption using a Cytation 3 plate reader (BioTek, Agilent Technologies, Santa Barbara, CA, USA) preheated to 37 °C and after equilibrating for 20 min. Lysis percentage in a well $n$ was calculated using the absorbance values $A$: Lysis (%) = 100 / (A$_{\text{total lysis}}$ − A$_{\text{spontaneous lysis}}$) ∗ (A$_n$ − A$_{\text{spontaneous lysis}}$).

## SRBC lysis measurements using haem release

As an alternative to turbidity-based measurements, SRBC lysis can be calculated based on colorimetric measurement of haem released by the ruptured cells. For these measurements, killing assays were set up the same way (see Methods) in 96-well plates. After the assay, the plate is centrifuged to pellet SRBCs, and 100 µL of supernatant is transferred to a flat bottom 96-well plate. The absorbance $A$ of each well at 410 nm was measured on a Cytation 3 plate reader and the lytic activity in a well $n$ calculated as: SRBC lysis (%) = 100/(A$_{\text{total lysis}}$ − A$_{\text{spontaneous lysis}}$) ∗ (A$_n$ − A$_{\text{spontaneous lysis}}$).

## Atomic force microscopy

AFM sample preparation and imaging largely followed previously published methods (Rudd-Schmidt et al, 2019). In brief, 4 µL of a 1 mg/mL suspension of unilamellar vesicles made from DOPC (Avanti Polar Lipids, Alabaster, AL, USA) were deposited on freshly cleaved mica (~10 mm diameter) covered in 100 µL MMT buffer, 25 mM MgCl$_2$, 5 mM CaCl$_2$, pH 7.4 and incubated for 30 min at room temperature until a supported lipid bilayer was formed. To adjust pH, the samples were washed 9 times with 80 µL MMT buffer, 25 mM MgCl$_2$, 5 mM CaCl$_2$, pH 4–6.5. Subsequently, WT-PRF was injected at a final concentration of ~150 nM and incubated for 5 min at 37 °C, and then imaged by AFM. To restore

pH to neutral, the samples were washed a further 9 times with 80 μL MMT buffer 25 mM MgCl$_2$, 5 mM CaCl$_2$, pH 7.4, incubated for 15 min at 37 °C, and re-imaged. AFM images were recorded on a MultiMode 8 system (Bruker, Santa Barbara, CA, USA) using PeakForce Tapping at 2 kHz with MSNL-E and -F probes (Bruker) at 50–100 pN of force. Images were processed in the NanoScope Analysis software (version 1.8, Bruker) for background flattening and colour (height) scale adjustments with the supported lipid bilayer surface set to 0 nm.

### WT-PRF fixation in AFM samples

Samples containing WT-PRF assemblies on a DOPC bilayer formed at pH 5 were incubated with 0.2% glutaraldehyde 8% solution (TAAB Laboratories Equipment, Aldermaston, Berks, United Kingdom) for 6 h at room temperature and subsequently washed three times with 80 μL MMT buffer, 25 mM MgCl$_2$, 5 mM CaCl$_2$, pH 5 before re-imaging.

### SRBC lysis recovery from acidic pH

WT-PRF dilution series were set up in 50 μL of MMT at pH 7.4 and pH 5.5 without Ca$^{2+}$ on ice and combined with 25 μL of dilute SRBCs ($\sim 6 \times 10^8$ cells/mL) in MMT containing 3 mM Ca$^{2+}$ per well of a 96-well flat-bottom plate. For SRBCs at pH 7.4, 125 μL of MMT pH 7.4, 1 mM Ca$^{2+}$ were added immediately, and the plate was incubated for 3 h or 6 h at 37 °C. To recover pH 5.5 to neutral pH, 125 μL of MMT pH 8.5, 1 mM Ca$^{2+}$, with or without 4 mM EGTA was added per well, gently mixed, and incubated for a further 30 min at 37 °C. To maintain pH 5.5, 125 μL of MMT pH 5.5, 1 mM Ca$^{2+}$ were added instead. For total lysis controls, 125 μL of water were added instead of buffer. Absorption of each well at 600 nm was measured on a Cytation 3 plate reader, and lysis calculated as outlined before.

### Mass photometry of perforin in solution

Stock solutions of WT-PRF or bovine serum albumin (BSA, Roche) in elution buffer, MMT pH 7.4, or MMT pH 5.5 at a concentration of 5 μg/mL were prepared. MMT buffers contained 1 mM CaCl$_2$. Mass photometry measurements were performed on a TwoMP system (Refeyn, Oxford, United Kingdom). The system was first focused with 16 μL of the respective buffer and subsequently injected with 4 μL of the stock solution to reach a final concentration of 1 μg/mL and the molecular weight data recorded. To restore WT-PRF in MMT pH 5.5 to neutral pH, MMT pH 8.5 was added to triple the sample volume. The system was then focused with 8 μL of buffer, and 12 μL of the restored sample was added to reach a final concentration of 1 μL/mL. All preparation and measurements were performed at room temperature.

### Antibody-based detection of purified TMH1-PRF

TMH1-PRF was incubated with SRBCs at a concentration of 1 μg/mL for 15 min on ice under the desired conditions. Cells were washed once and resuspended in RPMI media at neutral pH for antibody staining, as outlined under 'Antibodies'. Flow cytometry data for perforin

detection on SRBCs was analysed as shown in Appendix Fig. S5C. The cross-species reactivity and use of anti-human perforin clone δG9 to detect disulphide-locked murine perforin mutant TMH1-PRF was evaluated in Appendix Fig. S6.

### Electron microscopy data analysis

Negative-stain electron microscopy data from an earlier study (Lopez et al, 2013b) was generously provided by Helen Saibil and Natalya Lukoyanova of Birkbeck College, London, United Kingdom. The data were analysed on a user interface written in-house in Matlab (MathWorks, Natick, MA, USA) as used previously (Rudd-Schmidt et al, 2019). The generally well-resolved subunits of perforin assemblies in electron microscopy data allowed for manual marking of subunits to extract the number of subunits per assembly and inter-subunit distance, determined by linear interpolation.

### Cell line verification

Human-derived target cell lines U937 and K562 have been verified by small tandem repeat (STR) analysis shown in Appendix Table S2. STR profiles were recorded by the Australian Genome Research Facility (AGRF, Melbourne, Victoria, Australia). NK-92 effector cells were verified by flow cytometry-based phenotype analysis showing NKp44$^+$, Cd56$^+$, CD54$^+$, CD45RA$^+$, and CD16$^-$ expression (Appendix Fig. S7, CD16 not shown) in line with literature (Kotzur et al, 2022; Gong et al, 1994). Mouse-derived EL4 target cells were verified by flow cytometry-based analysis of the mCD90.2 phenotype found 100% positive compared to a human control cell line (data not shown). Cells were tested regularly for mycoplasma contamination by the Peter MacCallum Cancer Centre Genomics Core.

## Data availability

No primary datasets have been generated and deposited.

The source data of this paper are collected in the following database record: biostudies:S-SCDT-10_1038-S44319-024-00365-6.

## Peer review information

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

## Acknowledgements

The authors thank Helen Saibil and Natalya Lukoyanova (Birkbeck, University of London) for kindly providing electron microscopy data, for supporting this study, and critical reading of the manuscript; Annette Ciccone and Sandra Verschoor (Peter MacCallum Cancer Centre) for preparing purified perforin; Andrew Ellisdon (Monash University) for support of mass photometry experiments, Neil Young (University of Melbourne) for providing sheep red blood cells; Jane Oliaro (Peter MacCallum Cancer Centre) for providing the NK-92 cell line. Experiments were supported by the Peter MacCallum Cancer Centre Flow Cytometry and Genomics Cores, Media Kitchen, and Centre for Advanced Histology and Microscopy. This study was funded by the Swiss National Science Foundation p2skp3_187634 and P500PB_211089 awarded to AWH, the UK Medical Research Council MR/V009702/1 awarded to BWH and IV, and the Cancer Council Victoria and NHMRC 2011020 awarded to IV.

## Author contributions

**Adrian W Hodel**: Conceptualisation; Formal analysis; Funding acquisition; Validation; Investigation; Visualisation; Methodology; Writing—original draft; Project administration; Writing—review and editing. **Jesse A Rudd-Schmidt**: Formal analysis; Validation; Investigation; Visualisation; Writing—review and editing. **Tahereh Noori**: Formal analysis; Validation; Investigation; Visualisation; Writing—review and editing. **Christopher J Lupton**: Formal analysis; Validation; Investigation; Visualisation; Writing—review and editing. **Veronica C T Cheuk**: Formal analysis; Validation; Investigation; Visualisation; Writing—review and editing. **Joseph A Trapani**: Resources; Supervision; Writing—review and editing. **Bart W Hoogenboom**: Resources; Supervision; Funding acquisition; Project administration; Writing—review and editing. **Ilia Voskoboinik**: Conceptualisation; Resources; Supervision; Funding acquisition; Validation; Investigation; Writing—original draft; Project administration; Writing—review and editing.

Source data underlying figure panels in this paper may have individual authorship assigned. Where available, figure panel/source data authorship is listed in the following database record: biostudies:S-SCDT-10_1038-S44319-024-00365-6.

## Disclosure and competing interests statement

BWH holds an executive position at AFM manufacturer Nanosurf; Nanosurf played no role in the design and execution of this study. AWH, JRS, TN, CJL, VCC, JAT, and IV declare no competing interests.

# Expanded View Figures

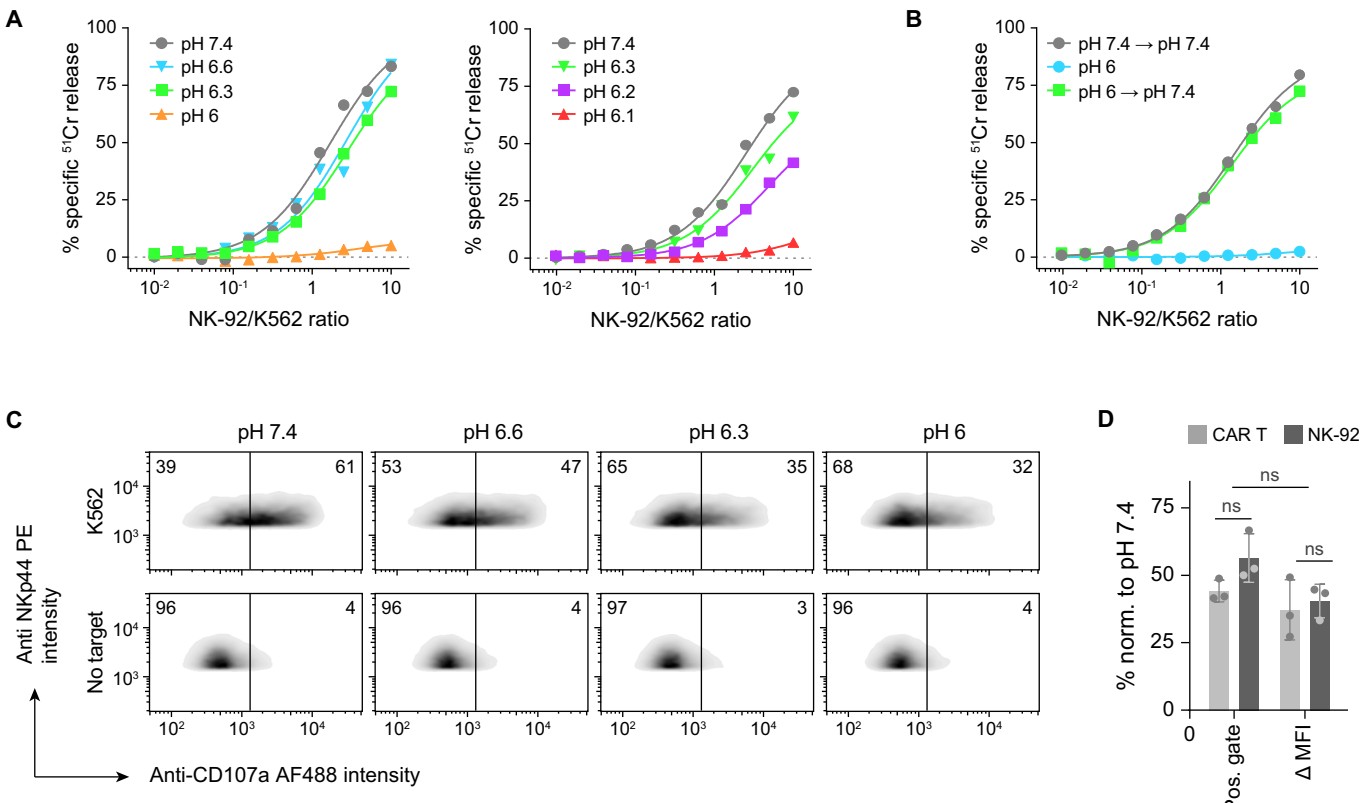

**Figure EV1. NK-92 cells degranulate but do not kill K562 target cells at acidic pH.**

(**A**) Killing assays performed at different pH outline a marked reduction of killing between pH 6.3–6. (**B**) Immune killing of cells incubated for 4 h at pH 6 and 37 °C was fully restored after overnight incubation at neutral pH, 37 °C. (**C**) Degranulation of NK-92 cells after mixing with K562 target cells at different pH shows a reduction, but not abrogation of degranulation. Of note, degranulation at pH 6.3 and 6 was almost identical, while the killing capacity was abolished (see (**A**)). (**D**) Degranulation at pH 6 compared to degranulation at neutral pH of CAR T (as in Fig. 1C) and NK-92 cells (as in (**C**)). Shown are degranulation levels estimated by positivity gate (pos. gate, as in (**C**)) or by increase of geometric mean fluorescence intensity compared to control samples without target cells (Δ MFI). Data information: (**A**) shows $n = 2$ biological replicates in separate plots. (**B**, **C**) shows data from single experiments, $n = 1$. In (**D**), data presents mean ± SD from $n = 3$ biological replicates. No significant differences between different conditions were found using Kolmogorov–Smirnov tests, ns, not significant, $P = 0.1$ (pos. gate), $P = 1.0$ (Δ MFI), $P = 0.5$ (pos. gate pooled vs. Δ MFI pooled).

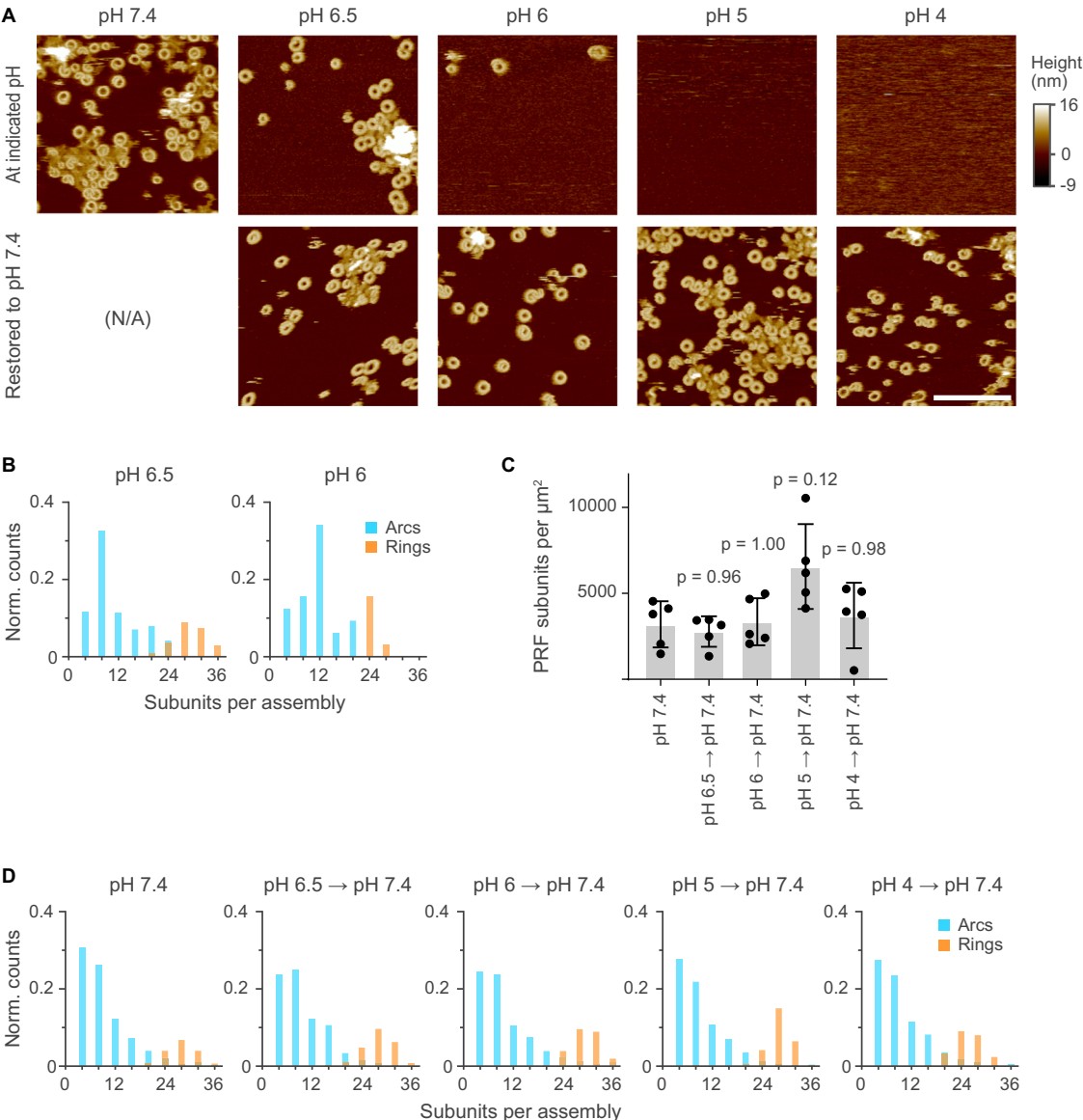

**Figure EV2. AFM detection of WT-PRF pores at different pH before and after neutralisation.**

(A) The top row of AFM images exemplifies pH-dependent WT-PRF pore formation. The samples were incubated with WT-PRF for 5 min at 37 °C and at indicated pH levels and subsequently imaged at room temperature for up to 1 h. The bottom row shows the same samples after restoring the pH by washing with pH 7.4 buffer and incubating for 15 min at 37 °C. The number of pores is visibly reduced at pH 6 before restoration and is absent in samples incubated at pH 5 and 4. After restoration of pH, WT-PRF pores emerge again in all samples, irrespective of prior pH. N/A, not assessed. (B) At pH 6.5 and pH 6, a reduced number of pores are formed. Their size distributions show arc- and ring-shaped assemblies. (C) An evaluation of protein densities on the sample surface after restoring pH from acidic conditions shows no significant deviations from samples prepared at neutral pH, showing that protein binding is not affected by pH. (D) Assembly size distributions of WT-PRF pores after their restoration to neutral pH. The shape of these distributions appears conserved across all pH levels. They also resemble distributions obtained on assemblies formed before the restoration of pH, shown in (B), overall indicating that the mechanism WT-PRF is not fundamentally altered by pH and can recommence its pore formation using the same oligomerization pathway as in neutral pH. The data shown in (B) and (D) was collected from 5 images taken across the sample surface, covering a total area of 1.5 μm² each. Data information: (A) scale bar, 200 nm. Distributions in (B) are shown from a total of $n = 212$ (pH 6.5) and $n = 32$ (pH 6) assemblies. In (C), Welch's ANOVA test with post hoc Dunnett's test detected no significant differences of protein densities of restored samples compared to pH 7.4; exact p values indicated atop each bar. Distributions in (D) are shown from a total of $n = 576$ (pH 7.4), $n = 417$ (pH 6.5 → pH 7.4), $n = 464$ (pH 6 → pH 7.4), $n = 965$ (pH 5 → pH 7.4), and $n = 593$ (pH 4 → pH 7.4) assemblies.

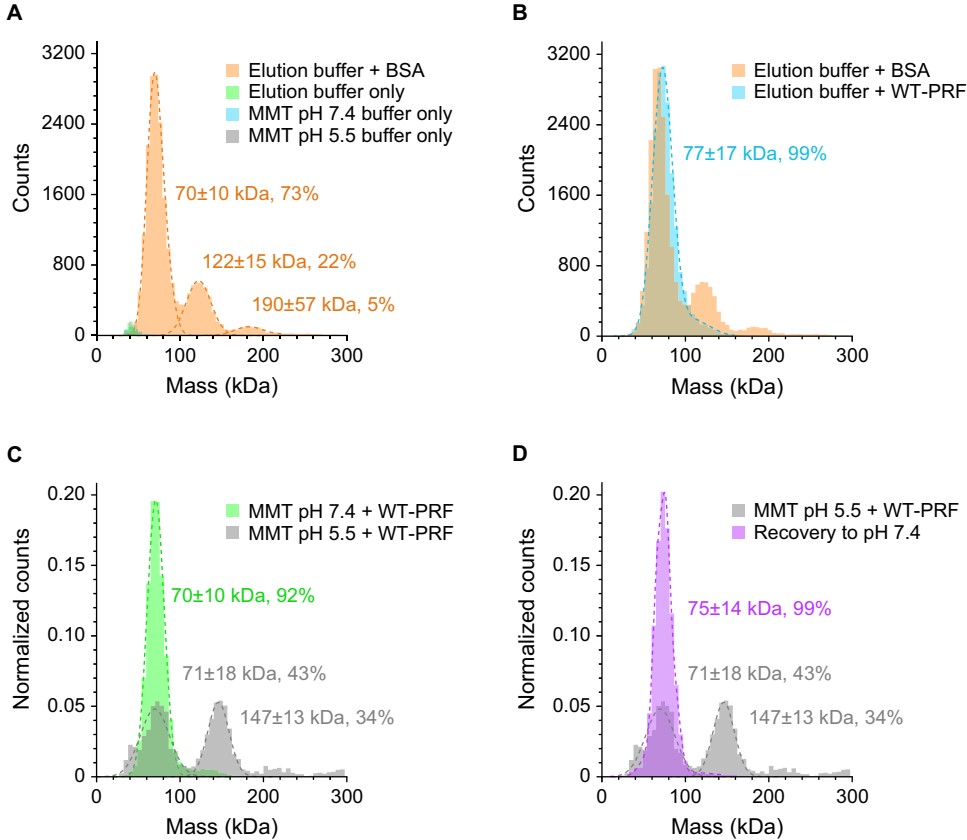

**Figure EV3. Mass photometry of WT-PRF in solution.**

(A) The prominent monomer and dimer peaks of bovine serum albumin (BSA, orange) in elution buffer were used to calibrate the mass photometry system. The buffers alone (green, blue, grey) produced a minor signal at around 40 kDa, which we interpret as noise at the lower boundary of the detection range. (B) Wild-type murine perforin (WT-PRF, blue) in elution buffer produced a single peak overlapping with the BSA (orange) monomer signal and estimated at 77 kDa molecular weight, approximately corresponding to the molecular mass of monomeric WT-PRF. (C) A similar singular peak for monomeric WT-PRF is observed in MMT buffer at pH 7.4 (green), whereas at pH 5.5, two peaks with molecular masses approximately corresponding to monomeric and dimeric WT-PRF are dominant (grey). (D) Dimeric WT-PRF found in MMT at pH 5.5 (grey, reproduced from (C)) disassembled into monomers upon restoring the buffer to pH 7.4 (purple).

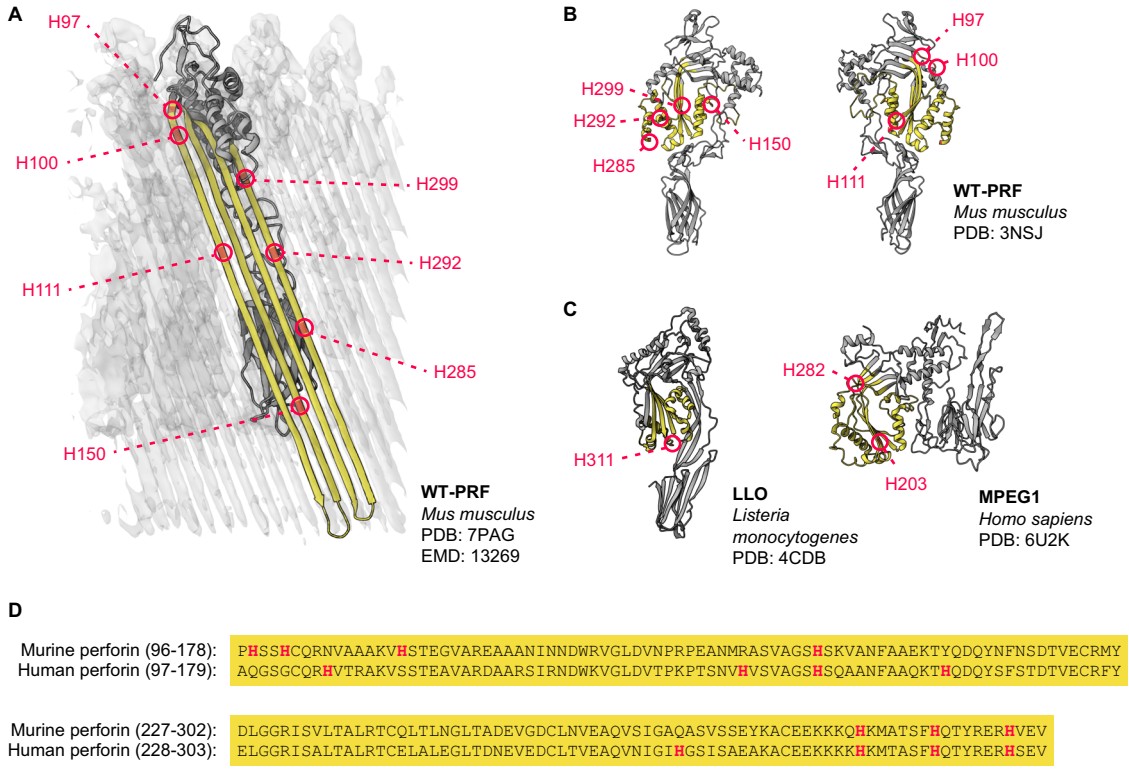

**Figure EV4. Histidine residues at transmembrane β-hairpin interfaces of perforin.**

(**A**) Cartoon of a murine WT-PRF subunit in the pore state, overlayed with a section of an electron density map of the perforin pore (grey) (Ivanova et al, 2022; Data ref: Ivanova et al, 2022a; Data ref: Ivanova et al, 2022b). The transmembrane β-barrel (yellow) motif of WT-PRF is seamed with numerous histidine residues (red). (**B**) The same histidine residues as in (**A**) highlighted in the soluble monomer structure of WT-PRF, shown from two sides (Law et al, 2010b; Data ref: Law et al, 2010b). (**C**) By comparison, the acid activated pore-forming proteins listeriolysin O (LLO) and macrophage expressed gene 1 (MPEG1) contain fewer histidine residues (red) in the corresponding regions (yellow, green). Protein models were generated from the RCSB protein database using the indicated model accessions (Köster et al, 2014; Pang et al, 2019; Data ref: Köster and Yildiz, 2014; Data ref: Pang and Bayly-Jones, 2019). (**D**) Comparison between wild-type murine and human perforin amino acid sequence for the β-barrel motif. The sequences were retrieved from UniProt under accessions P10820 and P14222, respectively, and aligned. Histidine residues are highlighted in red and are similarly abundant in both murine and human PRF, but not necessarily conserved.

