## [Peer Review File · EMBO Reports]

Acidic pH can attenuate immune killing through inactivation of perforin

Adrian Hodel, Jesse Rudd-Schmidt, Tahereh Noori, Christopher Lupton, Veronica Cheuk, Joseph A. Trapani, Bart Hoogenboom, and Ilia Voskoboinik

Corresponding authors: Adrian Hodel (adrian.hodel@petermac.org) , Ilia Voskoboinik (ilia.voskoboinik@petermac.org)

Review Timeline:

Submission Date:	7th Jul 24
Editorial Decision:	15th Aug 24
Revision Received:	13th Nov 24
Editorial Decision:	9th Dec 24
Revision Received:	17th Dec 24
Accepted:	19th Dec 24

Editor: Achim Breiling

Transaction Report:

Dear Dr. Hodel,

Thank you for the submission of your manuscript to EMBO reports. I have now received the reports from the three referees that were asked to evaluate your study, which can be found at the end of this email.

As you will see, the referees find the study interesting. However, they have several comments, concerns, and suggestions that need to be addressed to allow publication of the study in EMBO reports. As the reports are below, and all the concerns need to be addressed, I will not detail them further here.

Acceptance of your manuscript will depend on a positive outcome of a second round of review. It is EMBO reports policy to allow a single round of revision only and acceptance of the manuscript will therefore depend on the completeness of your responses included in the next, final version of the manuscript.

1) a .docx formatted version of the final manuscript text (including legends for main figures, EV figures and tables), but without the figures included. Figure legends should be compiled at the end of the manuscript text.

2) individual production quality figure files as .eps, .tif, .jpg (one file per figure), of main figures and EV figures. Please upload these as separate, individual files upon re-submission.

4) a complete author checklist, which you can download from our author guidelines

(<https://www.embopress.org/page/journal/14693178/authorguide>). Please insert page numbers in the checklist to indicate where the requested information can be found in the manuscript. The completed author checklist will also be part of the RPF.

5) that primary datasets produced in this study (e.g. RNA-seq, ChIP-seq, structural and array data) are deposited in an appropriate public database. If no primary datasets have been deposited, please also state this in a dedicated section (e.g. 'No primary datasets have been generated and deposited'), see below.

The accession numbers and database should be listed in a formal "Data Availability" section (placed after Materials & Methods) that follows the model below. This is now mandatory (like the COI statement). Please note that the Data Availability Section is restricted to new primary data that are part of this study. This section is mandatory. As indicated above, if no primary datasets have been deposited, please state this in this section

Data availability

8) Regarding data quantification and statistics, please make sure that the number "n" for how many independent experiments were performed, their nature (biological versus technical replicates), the bars and error bars (e.g. SEM, SD) and the test used to calculate p-values is indicated in the respective figure legends (also for EV figures and all those in an Appendix). Please also check that all the p-values are explained in the legend, and that these fit to those shown in the figure. Please provide statistical testing where applicable. Please avoid the phrase 'independent experiment', but clearly state if these were biological or technical replicates. Please also indicate (e.g. with n.s.) if testing was performed, but the differences are not significant. In case n=2, please show the data as separate datapoints without error bars and statistics. See also: <http://www.embopress.org/page/journal/14693178/authorguide#statisticalanalysis>

9) Please add scale bars of similar style and thickness to microscopic images, using clearly visible black or white bars (depending on the background). Please place these in the lower right corner of the images themselves. Please do not write on or near the bars in the image but define the size in the respective figure legend.

10) Please also note our reference format:

12) We now use CRediT to specify the contributions of each author in the journal submission system. CRediT replaces the author contribution section. Please use the free text box to provide more detailed descriptions and do NOT provide your final manuscript text file with an author contributions section. See also our guide to authors: <https://www.embopress.org/page/journal/14693178/authorguide#authorshipguidelines>

13) All Materials and Methods need to be described in the main text using our 'Structured Methods' format, which is required for all research articles. According to this format, the Materials and Methods section should include a Reagents and Tools Table (listing key reagents, experimental models, software, and relevant equipment and including their sources and relevant

identifiers), uploaded as separate file, followed by a Methods and Protocols section in which we encourage the authors to describe their methods using a step-by-step protocol format with bullet points, to facilitate the adoption of the methodologies across labs. More information on how to adhere to this format as well as downloadable templates (.doc) for the Reagents and Tools Table can be found in our author guidelines (section 'Structured Methods'):

14) Please order the manuscript sections like this, using these names:

Title page - Abstract - Keywords - Introduction - Results - Discussion - Methods - Data availability section - Acknowledgements including funding information) - Disclosure and Competing Interests Statement - References - Figure legends - Expanded View Figure legends

I look forward to seeing a revised form of your manuscript when it is ready.

Yours sincerely,

Referee #1:

Hodel et al, investigate the molecular mechanism explaining the loss of activity of perforin under acidic acid conditions. Perforin is a pore-forming protein, secreted into the immunological synapse to allow the entry of granzyme in target cells. Despite being explored as a potential molecule for immunotherapies against solid tumors, this strategy has been dampened due to the nature of the acidic conditions of the tumor microenvironment. They hypothesize that acidic pH can affect perforin function and test the effect of pH on immunological synapse and perform pore formation. They conclude that low pH affects perforin function specifically by preventing the formation of transmembrane β -barrel pores. While the hypothesis is appealing the conclusions require further validation with experimental data.

Major:

1. The effect of pH on perforin binding to the membrane should be tested. This can be done in the RBC system, in a liposome system, and also in the supported lipid bilayer system (based on the images already provided). Looking at the AFM images it is clear a reduction in arcs and rings (while their distribution remains unchanged) with pH. The number of structures at each pH can be quantified and included in the manuscript. This suggests that indeed pH is affecting insertion to membrane rather than pore assembly. The distribution of the size (radius and length) of these structures should also be included.
2. How does restoring the pH promote dimers dissociation and perforin assembly? Why do dimers would be ineffective for pore formation? Do they bind to membranes in a non-competent manner or dimerize in a configuration that is not suitable for further oligomerization? In other words, which kind of inactive dimers would be generated at low pH? Here a more detailed structural analysis and experimental validation are required. Indeed, it seems like the structures stabilized by the crosslinker in supplementary fig 9, represent protein aggregates. The rationale of the experiment shown in supplementary figs 9 and 10 should be better explained. Further mutations in the C2 domain and the calcium-binding site should be included, to demonstrate that perforin binds to lipids in acidic conditions through this domain (tested in membrane binding assays).
3. Are the structures obtained at low pH (as shown by EM) intermediate assemblies to the arc and rings? Or just inactive configurations? What is the difference in length distribution compared to the arcs? If they would be intermediate structures...shouldn't they be detected also at neutral pH? Why they cannot be visualized by AFM?
4. The model proposed in Figure 4 must be validated. Authors could design different point mutations (ex. Ala mutants) to assess the role of the proposed His in the mechanism of perforin assembly.

Minor:

1. The discussion can be expanded to include more comparisons with the existing literature and alternative models of perforin assembly and the implications of the study.
2. The number of supplementary figures can be reduced by merging panels contributing to the same main figures.

Referee #2:

Hodel et al. show that acidic pH levels reduce perforin pore formation and thus prevent cytotoxic lymphocyte killing of target cells. The authors present convincing evidence that the cytotoxicity of T and NK cells are compromised at pH less than 6.5 due to the inability of perforin to assemble into transmembrane pores. The experiments are well designed and comprehensive and support the conclusions of the paper.

My one comment - it does not seem common for tumors to have pH less than 6.5, at which the effects on perforin are observed. Do the authors believe that this has functional significance in vivo? Perhaps the effects on degranulation, which seems to be more prominent at the pH 6.5-7.4 range, is more physiologically relevant.

Minor comment - Figure 1 title does not accurately describe what the figure portrays (no mention of pH).

Referee #3:

In this manuscript the authors aimed to analyse how acidity impacts immune-mediated cytolysis from killer cells (CD8+ T and NK cells). They show that killing inhibition involves inactivation of perforin during its delivery to target cells.

Despite strong evidence in the literature that perforin is inactivated by low pH (which is well cited) and that pH dampens T-cell function (which would briefly need some description in the manuscript given the number of previous publications, especially the ones showing that killing is reduced at low pH: eg Nakagawa et al, Immunol. Lett. 2015; Vuillefroy de Silly et al, bioRxiv 2023), nothing was previously done to directly link, in an effector:target context/system, a dampened killing mediated by acidity and perforin inactivation. This study therefore shows that perforin inactivation should be considered while designing strategies to improve immunotherapy of cancer.

The manuscript is well written and most of the claims are adequately supported by experimental evidences. I still have two main comments.

Major comments:

1. My first concern is about the first claim of the authors: "Effector killing is attenuated at acidic pH despite sufficient release of perforin". To me, this claim ("sufficient release of perforin") is not fully supported experimentally. More specifically:
 - Data from Figure 1C and Supp Figure 2C: even if % positivity gives a valuable information, I believe that the authors should show the MFI. Indeed, positivity tells whether a cell has reached a fluorescence threshold. However, one might envision that one cell can have different level of degranulation, which could be estimated using the MFI (ideally on positive cells, since they are the one that display protein expression). Furthermore, there should be repetition of these experiments, as only one representative experiment is shown without results of pooled data.
 - Extrapolations made from Figure 1D, 1E and 1F: these graphs are indeed of interest, but they do not rule out a time-dependent effect that confound the interpretation. Maybe most of the CTLs at pH6 degranulated during the beginning of the assay (1-2 hours or less?), which led to CD107a positivity at the end of the assay but was not sufficient to kill target cells, and then acidity prevented degranulation during the rest of the assay. Actually, maybe the experiments with the release of ALFA-PRF could give an answer given that it looks ALFA-PRF signal was monitored over time. In line with this, the timing at which data from Supp Figure 3 are displayed is not provided, what is it?
 - It is not clear to me to which extent the cells and data from Noori et al used in Figure 1D can be comparable to those in Figure 1E and 1F (Cell type, species, timing, E:T ratio?). It would have been more direct, for example, to knock down one of the molecules involved in exocytosis/degranulation leading at least to the same degranulation extent as with pH6, and to show that at pH7.4 it is still leading to higher killing than at pH6.
2. My second main concern is linked to the title of the manuscript (" Acidic pH attenuates immune killing through inactivation of perforin"), which, in my opinion, misleadingly implies that killing inhibition is fully mediated by perforin inactivation. Indeed, the authors show that, while gradually acidifying the medium, there is a first phase where killing dampening could rather be degranulation rate -dependent and thus mostly perforin-inactivation -independent. It is mostly when reaching more acidic conditions (below 6.5) that stronger evidences for perforin-inactivation looks to be the main cause in the decreased killing efficiency observed.

Minor comments:

- Supp Fig 1 BCD -> n=1...
- What is the exact methodology (classical pH meter?) to measure pH in Supp Fig 1? Since typical carbonate-based media are

highly CO₂-dependent, simply taking out of the incubator the medium with some delay can increase a lot the pH artificially before the measurement.

- Line 80, the term "prolonged exposure" does not fit the reality in my opinion. 4 hours is rather a short/acute exposure (even if adequately done since the killing assay last 4 hours).
- Line 85, "we first tested antigen binding and degranulation", I missed the data or the authors did not test antigen binding. It could be inferred in case degranulation is the same because antigen binding leads to degranulation. However, even in this case, it is also possible that a slight decrease in antigen binding would not be sufficient to lower degranulation because the activation threshold is low enough.
- Line 87, "synapse formation", same comment as above: either I missed the data, or the authors did not assess synapse formation here.
- Line 90, "37% at pH6", but the data displayed show 32%, thus a 48% reduction instead.
- The importance of serglycin is not clear to me given that KO cells display equivalent rate of killing.
- Supp Fig9B: I am not familiar with these data, but what is supposed to indicate the blue dotted line?
- One may envision Figure 4 to be moved to Supplementary Figures since panels are hypothesis-driven illustrations without any experimental evidence. Its corresponding paragraph could also be moved/included in the discussion section.

We thank the Referees for their thorough study of our manuscript, and their insightful comments. Below are our responses in blue, and the references used in our responses are listed at the end of the document.

Referee #1:

Hodel et al, investigate the molecular mechanism explaining the loss of activity of perforin under acidic acid conditions. Perforin is a pore-forming protein, secreted into the immunological synapse to allow the entry of granzyme in target cells. Despite being explored as a potential molecule for immunotherapies against solid tumors, this strategy has been dampened due to the nature of the acidic conditions of the tumor microenvironment. They hypothesize that acidic pH can affect perforin function and test the effect of pH on immunological synapse and perform pore formation. They conclude that low pH affects perforin function specifically by preventing the formation of transmembrane β -barrel pores. While the hypothesis is appealing the conclusions require further validation with experimental data.

We thank the Referee for their comments and hope that our responses below will resolve the concerns.

Major:

1. The effect of pH on perforin binding to the membrane should be tested. This can be done in the RBC system, in a liposome system, and also in the supported lipid bilayer system (based on the images already provided). Looking at the AFM images it is clear a reduction in arcs and rings (while their distribution remains unchanged) with pH. The number of structures at each pH can be quantified and included in the manuscript. This suggests that indeed pH is affecting insertion to membrane rather than pore assembly. The distribution of the size (radius and length) of these structures should also be included.

Figure 3A shows similar TMH1-perforin binding to RBCs at pH 5.5 and pH 7.4, and we now included data of WT PRF binding to K562 cells (Figure EV4, reproduced below as Response Figure 1B) showing no significant difference of binding at pH 6 and pH 7.4.

Response Figure 1: Reproduction of Figure EV4 showing WT-PRF and (C2 mutant) D429A-PRF binding to K562 cells using flow cytometry. (A) K562 cells were treated with WT-PRF, D429A-PRF, or no protein (-PRF) at pH 7.4 (left plot) or pH 6 (right plot) and subsequently labelled with an anti-PRF antibody. Irrespective of pH, flow cytometry data shows increased fluorescence only in the presence of WT-PRF but not D429A-PRF, indicating that the C2 mutant is not binding K562 cells. (B) Grouped bar plot (mean \pm standard deviation of n=3 biological replicates) showing the

increase in geometric mean fluorescence intensity (Δ MFI) upon addition of WT-PRF and D429A-PRF at pH 7.4 and pH 6. Statistical significance was analysed using Kolmogorov-Smirnov tests.

As per the Referee's suggestion, the number of pore subunits in AFM images shown in Appendix Figure S3 was quantified in a new Figure panel (Appendix Figure S3C, reproduced below as Response Figure 2C). These samples were prepared at low pH, and then restored to neutral pH by replacing the supernatant and removing any unbound perforin. The number of perforin subunits present on the sample surface and, consequently, perforin binding is not significantly affected by pH as low as 4.

We have added the additional size distributions requested by the Referee in a new Appendix Figure S3B (reproduced below as Response Figure 2B) and included the Referee's interesting point in the Figure legends for Appendix Figure S3D. Of note, the number of pores formed at pH 6 is small, and no pores were detected at pH <6, despite the protein binding to the membrane. Therefore, we can only show size distributions (shown as numbers of subunits, as outlined in the manuscript) at pH 6.5 and 6. Radii (of curvature of oligomers) are not measured; these would be most meaningful in (short) arcs, where the tracing errors are largest, and we feel that this would not produce interpretable data at this point.

Response Figure 2: Reproduction of Appendix Figure S3C, panels B and C, providing quantification of AFM data. (B) At pH 6.5 and pH 6, a reduced number of pores are formed. Their size distributions show arc- and ring-shaped assemblies. (C) An evaluation of protein densities on the sample surface after restoring pH from acidic conditions shows no significant deviations from samples prepared at neutral pH, showing that protein binding is not affected by pH. No significant differences were found using Welch's ANOVA test with post hoc Dunnett's test (compared to pH 7.4, p values indicated atop each bar).

2. How does restoring the pH promote dimers dissociation and perforin assembly? Why do dimers would be ineffective for pore formation? Do they bind to membranes in a non-competent manner or dimerize in a configuration that is not suitable for further oligomerization? In other words, which kind of inactive dimers would be generated at low pH? Here a more detailed structural analysis and experimental validation are required. Indeed, it seems like the structures stabilized by the crosslinker in supplementary fig 9, represent protein aggregates. The rationale of the experiment shown in supplementary figs 9 and 10 should be better explained. [...]

We would love to better understand pH related effects on surface charge and protein-protein interaction and hope it will be possible to reliably compute this in the future. However, we currently do not know the nature of pH dependent perforin dimerization in solution.

With respect, we never claimed that perforin dimers are 'inactive' or unsuitable for further oligomerisation. In fact the opposite is true: as the binding data suggests (now better quantified as requested in the comment above), binding at low pH occurs just as much as at neutral pH, indicating that perforin dimers bind to the membrane as effectively as monomers. Once bound to the membrane, we observe prepore-like features at low pH, indicating that the assemblies consist of approximately five subunits on average (see Fig. 3D, bottom plot). Therefore, monomers and dimers bind the membrane surface and oligomerise further into prepores or prepore-like structures at acidic or neutral pH.

We have changed the wording in the legend of Appendix Figure S5 (former Sup. Figs. 9 & 10) to clarify the rationale and further reference (Leung *et al*, 2017), moved the reference forward where perforin prepores were introduced in the main text (line 61).

[...] Further mutations in the C2 domain and the calcium-binding site should be included, to demonstrate that perforin binds to lipids in acidic conditions through this domain (tested in membrane binding assays).

We show that perforin binds the lipid surface at acidic pH in a calcium dependent manner (Fig. 3A), which is a hallmark of the C2 domain, and in an orientation that suggests C2 is facing the correct way to interact with the substrate. Once perforin is bound at low pH, its activity can be fully restored upon neutralizing pH only in the presence of calcium (Figure 2D, Appendix Figure S3C), indicating that there is no aberrant binding interfering with pore formation.

Mutations of the critical Ca²⁺ binding residues in the perforin C2 domain lead to severely reduced coordination of calcium ions, thereby diminishing or abrogating its membrane-binding capacity. Several mutations are known that disrupt C2 function, and in particular the perforin C2 mutant D429A leads to a complete loss of perforin binding (Voskoboinik *et al*, 2005; Hodel *et al*, 2021; Yagi *et al*, 2015). As per the Referee's suggestion, we tested D429A binding to K562 cells at low pH and find that, as expected, it was absent; this is now shown in the new Figure EV4 (reproduced above as Response Figure 1) and explained in lines 180-182. This observation is fully consistent with our other observations that the C2 domain is required for perforin binding at acidic pH. Generation of other previously studied C2 domain mutants would require many months of work, and we hope that the Referee accepts the inclusion of only one mutant in response to their query.

3. Are the structures obtained at low pH (as shown by EM) intermediate assemblies to the arcs and rings? Or just inactive configurations? What is the difference in length distribution compared to the arcs? If they would be intermediate structures...shouldn't they be detected also at neutral pH? Why they cannot be visualized by AFM?

Our investigation of perforin deactivation at low pH follows the established mechanism of perforin pore formation at neutral pH (Leung *et al*, 2017). This mechanism constitutes distinct steps: perforin binding, oligomerization into short, non-lytic prepores, membrane insertion to form small pores, and lastly, further oligomerization in the membrane to mature into larger pores.

Prepores observed at neutral pH are indeed intermediate structures. (Leung *et al*, 2017) experimentally describe these intermediates using a disulphide-locked mutant protein that

cannot insert into the membrane (TMH1-PRF), and we consider these intermediates to be sufficiently established there.

At low pH, we observe short oligomers that are consistent with non-lytic prepores, as detailed in our manuscript and Figure 3B-D. They will remain trapped in such a state as long as the pH remains sufficiently acidic but can turn into pores when pH is neutralized - we refer to it as 'prepore-like' state.

The reason why prepores at neutral pH and prepore-like structures at low pH are not readily resolved with AFM is already established in (Leung *et al*, 2017), therein Figure 2 and Sup. Figures 5 & 6, which is now additionally cited in the legend of Appendix Figure S5 (former Sup. Fig. 9). In brief, since prepore intermediates are not membrane inserted, they are highly mobile and move laterally the membrane surface. In contrast, membrane inserted pores contact the solid sample substrate, rendering them static. Since AFM scanning is comparatively slow, it can resolve pores (static), but not prepores (dynamic), and the membrane may appear void of protein if only prepores are present. One established method to resolve prepores by AFM is to add a crosslinker such as glutaraldehyde, creating larger protein plaques that diffuse more slowly. We used these experimental properties to detect membrane bound perforin at low pH in Appendix Figure S5 (former Sup. Fig. 9). The related text sections have been improved as per our response to the previous comment.

4. The model proposed in Figure 4 must be validated. Authors could design different point mutations (ex. Ala mutants) to assess the role of the proposed His in the mechanism of perforin assembly.

To address the concern about the hypothetical nature of our model, we moved the former Figure 4 to the 'Extended View' data (now, Figure EV5) and the related text was moved to the Discussion section (lines 214-223).

We fully agree with the Referee that this model needs to be validated. To achieve this, several or all the 7-8 histidine residues identified in our model would need to be mutated. Some of these are identified in highly conserved regions and, potentially, such mutations may disrupt protein structure. Importantly, not only the histidine residues will need to be substituted, but also "pairing" residues that can form hydrogen bonds at acidic pH will need to be mutated, too. All these properties make it a difficult task to permutate pH resistant perforin. We believe that validating our hypothetical model and creating a pH resistant perforin mutant would warrant *in vivo* testing, which will require a separate research project.

Our aim was to establish whether acidic pH deactivates perforin within the immune synapse. We feel that we have demonstrated this and believe that our hypothesis is structurally sound, ending this manuscript with an uplifting look towards further possible research.

Minor:

1. The discussion can be expanded to include more comparisons with the existing literature and alternative models of perforin assembly and the implications of the study.

We added to the discussion to what extent the pH 6.5-6 range may be relevant for CTLs and, thus, immune therapies (lines 224-231). Further comparative literature has been added to the introduction as well (lines 54-56).

An alternative model that springs to mind proposes that perforin and granzymes are taken up via endocytosis, and that granzymes escape the endosome. This model was largely based on a lack of observation of membrane lesions that would indicate pore formation (Metkar *et al*, 2011). However, (Lopez *et al*, 2013) was able to visualize such lesions and has shown that in the physiological synapse, granzymes enter target cells directly via perforin pores faster than endosomal membrane repair is initiated and that, in addition, the rapid acidification of endosomes ought to deactivate endosomal perforin. We thus believe that it is sufficiently established that perforin/granzyme delivery via endocytosis is not a physiologically relevant process. However, we now cite historical reviews that suggest several potential modes of synergy between perforin and granzymes (Thiery & Lieberman, 2014; Gilbert *et al*, 2013; Voskoboinik *et al*, 2015) in the introduction (lines 34-35).

2. The number of supplementary figures can be reduced by merging panels contributing to the same main figures.

We merged former Sup. Fig. 4 with former Sup. Fig. 5 and former Sup Fig. 9 with former Sup. Fig. 10, creating the new Figure EV3 and Appendix Figure S5, respectively.

Referee #2:

Hodel et al. show that acidic pH levels reduce perforin pore formation and thus prevent cytotoxic lymphocyte killing of target cells. The authors present convincing evidence that the cytotoxicity of T and NK cells are compromised at pH less than 6.5 due to the inability of perforin to assemble into transmembrane pores. The experiments are well designed and comprehensive and support the conclusions of the paper.

My one comment - it does not seem common for tumors to have pH less than 6.5, at which the effects on perforin are observed. Do the authors believe that this has functional significance *in vivo*? Perhaps the effects on degranulation, which seems to be more prominent at the pH 6.5-7.4 range, is more physiologically relevant.

We thank the Referee for the opportunity to provide our subjective opinion in response.

Acidification is not only present in tumours but has also been shown in lymph nodes where certain areas equilibrate at around pH 6.3, caused by a high concentration of glycolytically active T cells (Wu *et al*, 2020). Tumours with their high rate of glycolysis and limited perfusion should be well able to reach similarly low pH.

Typically, tumour acidification has been thought to maintain pH above 6.5 in most cases. However, this contrasts with more recent *in vivo* pH sensitive MRI imaging, for which we referenced several papers in our manuscript. The images produced by such techniques show

heterogeneous (extracellular) pH distributions in tumour tissue frequently reaching pH 6-6.5 – including in human tumours (Jones *et al*, 2017).

We can only speculate why historic values differ from more recent pH measurement techniques, but we believe the extent of acidification may have been underestimated in the past. Perhaps it is due to measurements with relatively large probes that cannot resolve spatial variations in pH and measure some average instead, or that values have been reported as averages of multiple measurements in the first place. We also think that differences could be due to a rapid rise of pH if the measurements were taken after excision of the tumour; *in vivo*, tumour pH is buffered by a bicarbonate system (pKa ~6) and will depend on a high-CO₂ environment to maintain stable pH.

Furthermore, the pH values portrayed in MRI images are consistent with other measurement techniques. A recently introduced fluorescent probe consisting of a peptide that exclusively inserts the plasma membrane at pH <6.5 provided vivid images with widespread fluorescence signal in the periphery of tumours (Rohani *et al*, 2019). In an admittedly unconventional approach, (Miripour *et al*, 2020) assessed metastatic lymph node pH by probing tumour tissue with litmus paper *in vivo*, which produced values below pH 6 and some of the lowest values we have encountered so far. Such low pH values have recently also been established using nanometre sized, solubilized fluorescent probes that revealed polarized and highly acidic (pH 5.3) areas around cancer cells (Feng *et al*, 2024). This might be the most relevant study regarding your comment, as these highly acidic areas are most prominent at the plasma membrane surface where perforin unfolds its function. If immunological synapses would form indistinctively or even preferentially in such polarized low pH areas or avoid them is at present unclear. However, reports of immune co-receptor interactions involving VISTA specifically at pH ~6 (Yuan *et al*, 2021) support that effector-target interactions occur in such acidic environments.

In summary, it may be adequate to imagine a tumour as a heterogenous entity comprising zones of varying pH that can locally reach well below pH 6.5. CTLs need to infiltrate tumour tissue to unleash any activity against them, and thus they eventually encounter areas with sufficiently low pH that deactivates perforin. Furthermore, even a small volume of tumour tissue that resists immune killing is sufficient to prolong or sustain the disease. Lastly, however, our findings will need to be tested *in vivo*, possibly with CTLs carrying pH resistant perforin, for which we provide a starting hypothesis in terms of protein design.

We have added missing key aspects of this response to the introduction (lines 54-56) and discussion sections (line 224-231).

Minor comment - Figure 1 title does not accurately describe what the figure portrays (no mention of pH).

We amended the Figure Title to mention pH dependence (line 553).

Referee #3:

In this manuscript the authors aimed to analyse how acidity impacts immune-mediated cytotoxicity from killer cells (CD8+ T and NK cells). They show that killing inhibition involves inactivation of perforin during its delivery to target cells.

Despite strong evidence in the literature that perforin is inactivated by low pH (which is well cited) and that pH dampens T-cell function (which would briefly need some description in the manuscript given the number of previous publications, especially the ones showing that killing is reduced at low pH: eg Nakagawa et al, Immunol. Lett. 2015; Vuillefroy de Silly et al, bioRxiv 2023), nothing was previously done to directly link, in an effector:target context/system, a dampened killing mediated by acidity and perforin inactivation. This study therefore shows that perforin inactivation should be considered while designing strategies to improve immunotherapy of cancer.

The manuscript is well written and most of the claims are adequately supported by experimental evidences. I still have two main comments.

We thank the Referee for their feedback and have now included the two suggested references in the introduction and specifically mention reduced immune killing at low pH (lines 49-50).

Major comments:

1. My first concern is about the first claim of the authors: "Effector killing is attenuated at acidic pH despite sufficient release of perforin". To me, this claim ("sufficient release of perforin") is not fully supported experimentally. More specifically:

- Data from Figure 1C and Supp Figure 2C: even if % positivity gives a valuable information, I believe that the authors should show the MFI. Indeed, positivity tells whether a cell has reached a fluorescence threshold. However, one might envision that one cell can have different level of degranulation, which could be estimated using the MFI (ideally on positive cells, since they are the one that display protein expression). Furthermore, there should be repetition of these experiments, as only one representative experiment is shown without results of pooled data.

A new panel, Figure EV1D (reproduced below as Response Figure 3), now shows a biological triplicate of MFI and % positivity upon degranulation for CTLs and NK-92 at pH 6 and normalized to values observed at neutral pH. The average values are not significantly different when using MFI or % positivity, and the averages reside within 10-15% of each other, thus not affecting our conclusions.

Despite the overall good match between % positivity and MFI data for degranulation, the latter should be interpreted cautiously when comparing different cell types or mutants: each degranulation event increases MFI, but effectors may degranulate more than is required to kill a target cell. This is documented for NK cells (Gwalani & Orange, 2018) and is likely true for CTLs, too. Thus, a reduced MFI could still be above the threshold needed to kill a target cell, and there could arise seemingly paradoxical situations where MFI appears low while immune killing is high.

Response Figure 3: Reproduction of Figure EV1D. (D) $n=3$ biological triplicate values showing mean \pm standard deviation of degranulation at pH 6 compared to degranulation at neutral pH of CAR T (as in Figure 1C) and NK-92 cells (as in Figure EV1C). Shown are degranulation levels estimated by positivity gate (pos. gate, as in C) or by increase of geometric mean fluorescence intensity compared to control samples without target cells (Δ MFI). P values calculated using Kolmogorov-Smirnov tests are indicated.

- Extrapolations made from Figure 1D, 1E and 1F: these graphs are indeed of interest, but they do not rule out a time-dependent effect that confound the interpretation. Maybe most of the CTLs at pH6 degranulated during the beginning of the assay (1-2 hours or less?), which led to CD107a positivity at the end of the assay but was not sufficient to kill target cells, and then acidity prevented degranulation during the rest of the assay. Actually, maybe the experiments with the release of ALFA-PRF could give an answer given that it looks ALFA-PRF signal was monitored over time. In line with this, the timing at which data from Supp Figure 3 are displayed is not provided, what is it?

If perforin/granzymes are present and functional, degranulation always leads to immune killing. Even if degranulation events should be limited to earlier timepoints of the assay, we would still expect a proportional amount of immune killing. In addition, in our observations with timelapse light microscopy assays, the frequency of synapse formation at neutral pH will decrease over time as well, especially within the first hour of effector exposure to target cells.

Admittedly, we were also concerned that unforeseen time- and pH- dependent effects could introduce a bias into our degranulation assays. This could be, for example, a non-specific buildup of CD107 at the plasma membrane over time at low pH. We thus performed a time-course measurement of degranulation comparing neutral pH and pH 6, as shown below in Response Figure 4. In the absence of targets, we don't observe a pH dependent divergence of CD107 levels over time. In the presence of target cells, we observed an overall decrease of degranulation signal over time at pH 6 compared to neutral pH, indicating that degranulation measurements at 4 hours are conservative representations.

We have added the time of data acquisition ($t = 45$ min) to Figure EV2 (former Sup Fig. 3) in line 630.

Response Figure 4: Timelapse degranulation data. Left: CD107a mean fluorescence intensity (MFI) of anti-CD19 CAR T without target cells over time at pH 7.4 and pH 6. Right: Degranulation evaluated by positivity gate of anti-CD19 CAR T cells degranulating after exposure to U937 hCD19t target cells at pH 7.4 and pH 6. Degranulation data for 'pH 6 normalized to pH 7.4' is shown in addition in orange.

- It is not clear to me to which extent the cells and data from Noori et al used in Figure 1D can be comparable to those in Figure 1E and 1F (Cell type, species, timing, E:T ratio?). It would have been more direct, for example, to knock down one of the molecules involved in exocytosis/degranulation leading at least to the same degranulation extent as with pH6, and to show that at pH7.4 it is still leading to higher killing than at pH6.

Unfortunately, knocking out STXBP2 or MUNC13D involved in degranulation has a small effect on human T cells cytotoxicity and degranulation. Consequently, (Noori *et al*, 2023) assessed mouse T cells, where this was not an issue. Having said this, Figure 1A show the results of human T cells, and Figures 1G shows the results of mouse T cells. These are identical in their response to the reduced pH. Timing, cell types, E/T ratios for killing and degranulation are all following the same pattern. Hence, we utilised the results of (Noori *et al*, 2023), where mouse T cells were reconstituted with human proteins or their mutants. The results in this paper demonstrate that reconstituted mutant proteins with severely reduced, yet measurable, degranulation capacity still exhibit some cytotoxic activity, and the patient presents with an atypical disease. In other words, residual degranulation is consistently associated with cytotoxicity. However, in our case, despite significant (though reduced) degranulation at acidic pH (Figure 1), the cells show no cytotoxic activity. We felt it was important to present Noori et al.'s data as a positive control.

2. My second main concern is linked to the title of the manuscript (" Acidic pH attenuates immune killing through inactivation of perforin"), which, in my opinion, misleadingly implies that killing inhibition is fully mediated by perforin inactivation. Indeed, the authors show that, while gradually acidifying the medium, there is a first phase where killing dampening could rather be degranulation rate -dependent and thus mostly perforin-inactivation -independent. It is mostly when reaching more acidic conditions (below 6.5) that stronger evidences for perforin-inactivation looks to be the main cause in the decreased killing efficiency observed.

We changed the title from 'Acidic pH attenuates immune killing through inactivation of perforin' to 'Acidic pH can attenuate immune killing through inactivation of perforin'. We also improved the Introduction (lines 54-56) and Discussion (lines 224-231) sections to better outline that the occurrence of levels below pH 6.5 around cancer cells has been observed in various settings and is realistic.

Minor comments:

- Supp Fig 1 BCD -> n=1...

The purpose of this Figure is to show pH stability and cell viability of/in our pH stable media. Panels A and B are replicates of C and D using a different biological system (human vs. mouse effectors), representing n=2 datasets showing pH stability and cell viability.

To show further data, we have pooled two conditions using FBS and BSA in the media in panels C and D and outlined this in the captions. FBS and BSA are functionally equivalent for the purpose of these experiments.

- What is the exact methodology (classical pH meter?) to measure pH in Supp Fig 1? Since typical carbonate-based media are highly CO₂-dependent, simply taking out of the incubator the medium with some delay can increase a lot the pH artificially before the measurement.

The Referee is absolutely correct, which is why we locked the (classical) pH probe inside the incubator during the measurements. We have now added this to the Figure legend in Appendix Figure S1.

- Line 80, the term "prolonged exposure" does not fit the reality in my opinion. 4 hours is rather a short/acute exposure (even if adequately done since the killing assay last 4 hours).

We changed 'the prolonged exposure' to 'the 4 h exposure' (lines 89 and 124).

- Line 85, "we first tested antigen binding and degranulation", I missed the data or the authors did not test antigen binding. It could be inferred in case degranulation is the same because antigen binding leads to degranulation. However, even in this case, it is also possible that a slight decrease in antigen binding would not be sufficient to lower degranulation because the activation threshold is low enough.

We removed the inferral of antigen binding from this statement (lines 94-95), as it seemed redundant after introducing the signalling cascade leading to degranulation just in the sentence prior.

- Line 87, "synapse formation", same comment as above: either I missed the data, or the authors did not assess synapse formation here.

We rephrased the sentence in lines 96-98 more carefully to specify that by confirming (reduced, but present) degranulation, we simultaneously infer (possibly reduced, but present) synapse formation.

- Line 90, "37% at pH6", but the data displayed show 32%, thus a 48% reduction instead.

We apologize for the mistake and revise the percentage in our manuscript (line 100).

- The importance of serglycin is not clear to me given that KO cells display equivalent rate of killing.

The referee is correct in pointing out the equivalent rate of killing of wild-type and serglycin KO immune effectors, from which we infer that serglycin plays no role in attenuating immune killing at acidic pH (lines 126-127). We have rewritten lines 119-122 to better outline the (potential) role of serglycin in perforin mediated immune killing and the rationale why the influence of serglycin must be assessed:

"Serglycin sequesters and thereby inactivates perforin at the low pH in cytotoxic granules during storage and, after secretion, dissociates at the neutral pH of the immunological synapse. To assess if serglycin plays a role in attenuating immune killing at acidic pH, we tested whether serglycin deficiency improves immune killing in OTI CTLs."

- Supp Fig9B: I am not familiar with these data, but what is supposed to indicate the blue dotted line?

The dotted line in Appendix Figure S5B (former Sup. Fig. 9B) indicates where the height-profile in Appendix Figure S5C (former Sup. Fig. 9C) was taken from. We now included the explanation in the Figure legend for panel B.

- One may envision Figure 4 to be moved to Supplementary Figures since panels are hypothesis-driven illustrations without any experimental evidence. Its corresponding paragraph could also be moved/included in the discussion section.

We followed this Referee's and Referee 1's suggestions and moved Figure 4 into 'Extended View' (Figure EV5) and related results text into the Discussion section (lines 214-223).

References

- Feng Q, Bennett Z, Grichuk A, Pantoja R, Huang T, Faubert B, Huang G, Chen M, DeBerardinis RJ, Sumer BD, *et al* (2024) Severely polarized extracellular acidity around tumour cells. *Nature Biomedical Engineering* 2024 8:6 8: 787–799
- Gilbert RJC, Mikelj M, Dalla Serra M, Froelich CJ & Anderlueh G (2013) Effects of MACPF/CDC proteins on lipid membranes. *Cell Mol Life Sci* 70: 2083–2098
- Gwalani LA & Orange JS (2018) Single Degranulations in NK Cells Can Mediate Target Cell Killing. *The Journal of Immunology* 200: 3231–3243
- Hodel AW, Rudd-Schmidt JA, Trapani JA, Voskoboinik I & Hoogenboom BW (2021) Lipid specificity of the immune effector perforin. *Faraday Discuss* 232: 236–255
- Jones KM, Randtke EA, Yoshimaru ES, Howison CM, Chalasani P, Klein RR, Chambers SK, Kuo PH & Pagel MD (2017) Clinical Translation of Tumor Acidosis Measurements with AcidoCEST MRI. *Mol Imaging Biol* 19: 617–625
- Leung C, Hodel AW, Brennan AJ, Lukoyanova N, Tran S, House CM, Kondos SC, Whisstock JC, Dunstone MA, Trapani JA, *et al* (2017) Real-time visualization of perforin nanopore assembly. *Nat Nanotechnol* 12: 467–473
- Lopez JA, Susanto O, Jenkins MR, Lukoyanova N, Sutton VR, Law RHP, Johnston A, Bird CH, Bird PI, Whisstock JC, *et al* (2013) Perforin forms transient pores on the target cell plasma membrane to facilitate rapid access of granzymes during killer cell attack. *Blood* 121: 2659–2668
- Metkar SS, Wang B, Catalan E, Anderlueh G, Gilbert RJC, Pardo J & Froelich CJ (2011) Perforin Rapidly Induces Plasma Membrane Phospholipid Flip-Flop. *PLoS One* 6: e24286
- Miripour ZS, Aghaee P, Abbasvandi F, Hoseinpour P, Parniani M & Abdolahad M (2020) Real-time diagnosis of sentinel lymph nodes involved to breast cancer based on pH sensing through lipid synthesis of those cells. *Biosci Rep* 40: 20200970
- Noori T, Rudd-Schmidt JA, Kane A, Frith K, Gray PE, Hu H, Hsu D, Chung CWT, Hodel AW, Trapani JA, *et al* (2023) A cell-based functional assay that accurately links genotype to phenotype in familial HLH. *Blood* 141: 2330–2342
- Rohani N, Hao L, Alexis MS, Joughin BA, Krismer K, Moufarrej MN, Soltis AR, Lauffenburger DA, Yaffe MB, Burge CB, *et al* (2019) Acidification of Tumor at Stromal Boundaries Drives Transcriptome Alterations Associated with Aggressive Phenotypes. *Cancer Res* 79: 1952–1966
- Thiery J & Lieberman J (2014) Perforin: a key pore-forming protein for immune control of viruses and cancer. *Subcell Biochem* 80: 197–220
- Voskoboinik I, Thia M-C, Fletcher J, Ciccone A, Browne K, Smyth MJ & Trapani JA (2005) Calcium-dependent plasma membrane binding and cell lysis by perforin are mediated through its C2 domain: A critical role for aspartate residues 429, 435, 483, and 485 but not 491. *Journal of Biological Chemistry* 280: 8426–8434

- Voskoboinik I, Whisstock JC & Trapani JA (2015) Perforin and granzymes: function, dysfunction and human pathology. *Nature Reviews Immunology* 2015 15:6 15: 388–400
- Wu H, Estrella V, Beatty M, Abrahams D, El-Kenawi A, Russell S, Ibrahim-Hashim A, Longo DL, Reshetnyak YK, Moshnikova A, *et al* (2020) T-cells produce acidic niches in lymph nodes to suppress their own effector functions. *Nat Commun* 11: 4113
- Yagi H, Conroy PJ, Leung EWW, Law RHP, Trapani JA, Voskoboinik I, Whisstock JC & Norton RS (2015) Structural Basis for Ca²⁺-mediated Interaction of the Perforin C2 Domain with Lipid Membranes. *Journal of Biological Chemistry* 290: 25213–25226
- Yuan L, Tatineni J, Mahoney KM & Freeman GJ (2021) VISTA: A Mediator of Quiescence and a Promising Target in Cancer Immunotherapy. *Trends Immunol* 42: 209–227

Dear Dr. Hodel,

Thank you for the submission of your revised manuscript to our editorial offices. I have now received the reports from two referees that I asked to re-evaluate the study, you will find below. As you will see, both referees now fully support the publication of the study in EMBO reports. Both referees have suggestions to improve the study, I ask you to address in a final revised manuscript.

Moreover, I have these editorial requests I also ask you to address:

- We plan to publish your manuscript as a Report. However, for a Scientific Report we require that results and discussion sections are combined in a single chapter called "Results & Discussion". Please do this for your manuscript. For more details please refer to our guide to authors: <http://www.embopress.org/page/journal/14693178/authorguide#researcharticleguide>
 - A report can have up to 5 main figures. Looking at the present figures, I note, as also indicated by referee #1, that there is an unbalance between main and supplementary figures. I think it would be no problem to show the content of the present three main figures and the 5 small EV figures in 5 new main figures. Please do that. You might also consider to then show some of the Appendix figures as EV figures. After the changes done, please update the legends and all callouts. Important: We request source data for all the main figures. Please also update the source data for the new 5 main figures.
 - We do not allow supplementary methods. All methods and reference information needs to be provided in the main manuscript text. Please move both from the Appendix to the main manuscript text file.
 - Please make sure that all figure panels (main, EV and Appendix figures) are called out separately and sequentially. Presently, there are no callouts for Appendix Figures S7-S9, and Appendix Table S2. Please check.
 - Please add scale bars of similar style and thickness to all microscopic images (presently shown only in the Appendix, it seems), using clearly visible black or white bars (depending on the background). Please place these in the lower right corner of the images themselves. Please do not write on or near the bars in the image but define the size in the respective figure legend. Presently, scale bars are missing in Fig. EV2B.
 - Please check that the number "n" for how many independent experiments were performed, their nature (biological versus technical replicates), the bars and error bars (e.g. SEM, SD) and the test used to calculate p-values is indicated in the respective figure legends (main and EV figures). Please also check that all the p-values are explained in the legend, and that these fit to those shown in the figure. Please provide statistical testing where applicable. Please avoid the phrase 'independent experiment', but clearly state if these were biological or technical replicates. Please also indicate (e.g. with n.s.) if testing was performed, but the differences are not significant. In case n=2, please show the data as separate datapoints without error bars and statistics. See also:
<http://www.embopress.org/page/journal/14693178/authorguide#statisticalanalysis>
- If n<5, please show single datapoints for diagrams. Could statistics be added to the diagram in Appendix Fig. S1A? Moreover:
- Please note that the exact p values are not provided in the legends of figures 1h; 3c; EV 2a.
 - Please note that information related to n is missing in the legend of figure 2c.
 - Please note that the error bars are not defined in the legend of figure 2c.
 - Please note that scale bar and its definition are missing for figure EV 2b.
- Please add to each legend (main, EV figures, Appendix figures where applicable) a 'Data Information' section explaining the statistics used or providing information regarding replicates and scales. See:
<https://www.embopress.org/page/journal/14693178/authorguide#figureformat>
 - Please add a table of contents to the Appendix mentioning each item with its corresponding page number.
 - Please make sure that all the funding information is also entered into the online submission system and is complete and similar to the one in the manuscript text file (in the Acknowledgements). Presently, the funders Peter MacCallum Cancer Centre Flow Cytometry and Genomics Cores, Media Kitchen, and Centre for Advanced Histology and Microscopy are missing from the submission system. Please check.
 - The panels shown in Fig. 2B are all also shown in Fig. S3A and partly in S5A. Please clearly mention and explain this re-sue in the respective figure legends.

In addition, I would need from you uploaded separately:

- a short, two-sentence summary of the manuscript (not more than 35 words).
- two to four short (!) bullet points highlighting the key findings of your study (two lines each).

- a schematic summary figure as separate file that provides a sketch of the major findings (not a data image) in jpeg or tiff format (with the exact width of 550 pixels and a height of not more than 400 pixels) that can be used as a visual synopsis on our website.

Best,

Referee #1:

The manuscript has notably improved. There is some unbalance between main and supplementary figures. I recommend to move to the main section some of the supplementary figures as the manuscript only has 3 main ones.

Referee #3:

The authors mostly answered positively to my concerns. I still have this comment:
I still believe the title of the first part claiming « Effector killing is attenuated at acidic pH despite sufficient release of perforin » is not supported by the data. In particular, this sentence implies that the perforin released at acidic pH should be sufficient to obtain the same levels of killing than at neutral pH. In response to my concern, the authors replied: "effectors may degranulate more than is required to kill a target cell. This is documented for NK cell and is likely true for CTLs, too. Thus, a reduced MFI could still be above the threshold needed to kill a target cell, and there could arise seemingly paradoxical situations where MFI appears low while immune killing is high.": indeed, but the authors show no proof supporting "sufficient" release of perforin (/degranulation) so that to lead to the same killing as at neutral pH... Instead it is rather the opposite that the authors are showing: the number of degranulating cells is decreased and the degranulation events per cell are lowered. One may consider subtly changing the section title (eg "Effector killing is attenuated at acidic pH despite significant release of perforin"). Minor comment: the second reference advised, that has now been included by the authors, is no longer a preprint (published in the EMBO J).

We thank the Referees for finding our responses satisfactory and for their continued valuable input. Below are our responses, highlighted in blue.

Referee #1:

The manuscript has notably improved. There is some unbalance between main and supplementary figures. I recommend to move to the main section some of the supplementary figures as the manuscript only has 3 main ones.

We have now increased the number of main and EV figures in line with the Referee's and the Editor's suggestions.

Referee #3:

The authors mostly answered positively to my concerns. I still have this comment: I still believe the title of the first part claiming « Effector killing is attenuated at acidic pH despite sufficient release of perforin » is not supported by the data. In particular, this sentence implies that the perforin released at acidic pH should be sufficient to obtain the same levels of killing than at neutral pH. In response to my concern, the authors replied: "effectors may degranulate more than is required to kill a target cell. This is documented for NK cell and is likely true for CTLs, too. Thus, a reduced MFI could still be above the threshold needed to kill a target cell, and there could arise seemingly paradoxical situations where MFI appears low while immune killing is high.": indeed, but the authors show no proof supporting "sufficient" release of perforin (/degranulation) so that to lead to the same killing as at neutral pH... Instead it is rather the opposite that the authors are showing: the number of degranulating cells is decreased and the degranulation events per cell are lowered. One may consider subtly changing the section title (eg "Effector killing is attenuated at acidic pH despite significant release of perforin"). Minor comment: the second reference advised, that has now been included by the authors, is no longer a preprint (published in the EMBO J).

The Referee is correct, and we did not intend to suggest that reduced degranulation has no effect on immune killing. Rather, we meant that the reduction in degranulation alone cannot explain the observed attenuation of immune killing. We thank the Referee for suggesting a better title, and we have made the change in the manuscript accordingly.

The reference for Vuillefroy de Silly et al. is now updated.

Dr. Adrian Hodel
Peter MacCallum Cancer Centre
305 Grattan St
Victoria 3052
Australia

Dear Dr. Hodel,

I am pleased to inform you that your manuscript has been accepted for publication in EMBO reports. Your manuscript will be processed for publication by EMBO Press. It will be copy edited and you will receive page proofs prior to publication. Please note that you will be contacted by Springer Nature Author Services to complete licensing and payment information.

Yours sincerely,
